# Gut Microbiome Dysbiosis as a Potential Risk Factor for Idiopathic Toe-Walking in Children: A Review

**DOI:** 10.3390/ijms241713204

**Published:** 2023-08-25

**Authors:** Svetlana Kononova, Mikhail Kashparov, Wenyu Xue, Natalia Bobkova, Sergey Leonov, Nikolaj Zagorodny

**Affiliations:** 1Institute of Protein Research, Russian Academy of Sciences, 142290 Pushchino, Russia; 2Department of Traumatology and Orthopedics, Peoples’ Friendship University of Russia, 117198 Moscow, Russia; kashparovmikhail@gmail.com (M.K.); zagorodniy51@mail.ru (N.Z.); 3Scientific and Practical Center for Child Psychoneurology, 119602 Moscow, Russia; 4School of Biological and Medical Physics, Moscow Institute of Physics and Technology, 141700 Dolgoprudny, Russia; syue.v@phystech.edu (W.X.); leonov.sv@mipt.ru (S.L.); 5Institute of Cell Biophysics, Russian Academy of Sciences, 142290 Pushchino, Russia; nbobkova@mail.ru; 6N.N. Priorov Central Research Institute of Traumatology and Orthopedics, 127299 Moscow, Russia

**Keywords:** gut microbiota dysbiosis, idiopathic toe walking, microbiota–gut–brain axis, neurotransmitter dysbalance, sensory–motor dysfunction

## Abstract

Idiopathic toe walking (ITW) occurs in about 5% of children. Orthopedic treatment of ITW is complicated by the lack of a known etiology. Only half of the conservative and surgical methods of treatment give a stable positive result of normalizing gait. Available data indicate that the disease is heterogeneous and multifactorial. Recently, some children with ITW have been found to have genetic variants of mutations that can lead to the development of toe walking. At the same time, some children show sensorimotor impairment, but these studies are very limited. Sensorimotor dysfunction could potentially arise from an imbalanced production of neurotransmitters that play a crucial role in motor control. Using the data obtained in the studies of several pathologies manifested by the association of sensory–motor dysfunction and intestinal dysbiosis, we attempt to substantiate the notion that malfunction of neurotransmitter production is caused by the imbalance of gut microbiota metabolites as a result of dysbiosis. This review delves into the exciting possibility of a connection between variations in the microbiome and ITW. The purpose of this review is to establish a strong theoretical foundation and highlight the benefits of further exploring the possible connection between alterations in the microbiome and TW for further studies of ITW etiology.

## 1. Introduction

Psychomotor development refers to age-related motor changes occurring during a person’s lifetime related to their interactions with the environment and surroundings [1,2]. During infancy, children experience a series of motor milestones (crawling, sitting and standing, upright posture, walking, etc.) with the acquisition and improvement of postural control. Anatomy and physiology change dramatically with the growth of the child [3]. Until about 7–8 years of age, children’s walking ability gradually develops from the initial independent walking, providing experiences during both each step and fall [4,5]. Sometimes toddlers between the ages of one and a half and two walk on their toes. When a child’s initial foot contact is on the forefoot instead of the common heel strike, it is referred to as toe walking (TW). It can result from normal development, harmless issues, or even severe conditions. TW severity can range from typical childhood development to congenital or acquired contractures, as well as neurological or neuromuscular disorders that impact muscle function. Nearly 59% and 80% of children who toe walk as a phase of normal gait development during early development [6] recover normal walking (a heel–toe gait) by the age of 5.5 [7] and 10 [8], respectively. However, there are cases in which TW persists in children, which might be unilateral or even develop after a period of heel walking [9]. If healthy children without signs of neurological, orthopaedic, or psychiatric diseases exhibit persistent TW after they should typically have achieved a heel–toe gait, they are diagnosed with idiopathic toe walking (ITW). ITW is a diagnosis of exclusion [10,11,12]. Generally, estimates of ITW prevalence vary widely from different studies: from 2% of children by Engström and Tedroff [7] to 7–24% by Furrer [13] and up to 12% by Engelbert [10].

Although ITW has been known for more than half a century, we still do not know much about its etiology [14,15]. Currently, there is a growing interest in the diagnosis, treatment, classification, and causes of ITW. This is due, in part, to the fact that conservative treatment of persistent ITW often yields unsustainable results [16,17,18]. The considerable increase in the volume of literature pertaining to this subject over the last five years is a compelling indication of the mounting interest in this field. These publications represent more than one-third of all articles currently published on PubMed, highlighting the significance of this subject in the scientific community.

The prevailing TW history among families with ITW strongly suggests the involvement of genetic factors in the manifestation of this condition within specific individuals [7,19,20,21]. Indeed, Pomarino and his colleagues propose that genetic abnormalities may be the root cause of the frequent occurrence of ITW in children during their developmental stage [21,22,23]. Several mutations were found in addition to defects in the 129 genes of uncertain significance. These mutations may lead directly to toe walking lesions and gait abnormalities [23]. Of note, children who were found to have mutations that led to the development of ITW did not experience any birth-related complications before, during, or immediately after their birth [22,23].

Some research studies have highlighted the presence of altered sensory processing, encompassing tactile, proprioceptive, vestibular, and visual processing, in several children affected by ITW. Furthermore, several initiatives were implemented to confirm the disruption of sensorimotor regulation [24,25,26,27,28,29,30,31].

Studies have indicated a significant link between ITW in children, their family’s history, and sensory processing difficulties, along with autism/autism spectrum disorders (ASD), attention-deficit/hyperactivity disorder (ADHD), and speech development disorders [32,33,34]. In these circumstances, there arises the question of whether this diagnosis is truly “idiopathic” or if there is a genuine underlying neurological factor involved [28,34].

These findings suggest that the severity and clinical presentation of ITW may depend on a combination of mutated variants and various risk factors, such as perinatal factors [23,34,35], which may affect the development of sensorimotor impairment. Of note, showing the merit of continued genetic research into the genetic causes of ITW, the research has the potential to identify ITW as a symptom for further conditions, which it was not previously linked to.

In a recent review, Bauer et al. [15] proposed a new nomenclature for ITW. The children in the ITW study were categorized depending on their age and the type of toe walking they exhibited. The identified categories comprise the following: children below 5 years of age, who walk on their toes without contractures; older children who have a persistent habit of walking on their toes; and children who exhibit toe walking due to congenital or acquired contractures. ITW has been classified into four distinct forms, namely, “Developmental ITW”, “ITW + ASD”, “Congenital contracture”, and “Persistent ITW”. The last form consists of two subgroups labeled as “Contracture” and “Habitual”. In their opinion, such an approach should help distinguish between forms of ITW and build a better treatment strategy.

From the nomenclature proposed by Bauer et al., it follows that sensorimotor disorders are associated with ASD [15]. The categorization of “ITW + ASD” disagrees with the term ITW [10,11,12]. Hence, talking about the group “TW + ASD” but leaving it off the ITW classification would be a good idea. ASD belongs to a group of neurodevelopmental disorders [36] that can manifest with sensorimotor impairments. However, “TW + ASD” is actually very significant because it highlights the well-known connection between ASD and an imbalance in the neurotransmitter system that is caused by dysbiosis of the gut microbiome [37,38,39,40]. Revisiting Bauer et al.’s proposed ITW classification [15] through the lens of genetic disorders and subclinical conditions in sensorimotor control, we suggest that an imbalance in metabolic regulation of neurotransmitter synthesis due to dysbiosis of the gut microbiota may account for “Developmental ITW” and partially for “Persistent ITW” (Figure 1).

There is overwhelming evidence to support the similar mechanism of the significant impact of gut microbiota in the development of various neurodevelopmental and neurodegenerative conditions [41,42,43,44,45,46]. That may serve as indirect confirmation of the relevance of such an assumption.

For the first time, in this review, we suggest that disturbances of neurotransmitter production as a result of dysbiosis of the intestinal microbiota at the age of 2–3 years may underlie the expression of the ITW phenotype in the “Developmental” and “Persistent” ITW groups and not only in the “TW + ASD” group. We also pose that the impact of dysbiosis in the “TW + ASD” group might be much more extensive, affecting not only sensory processing but also communication and social interaction [36].

This review explores the potential link between microbiome variations and ITW. Our belief is that validated assumptions provide hope for treating the ITW symptoms in the initial stage, before irreversible morphological lesions occur.

## 2. Mechanics of Walking

Early childhood is a key stage of development for kids, during which critical progress in motor, psychological, and cognitive health development closely connects with physical activity [47]. Motor development addresses the development of anatomical, neuromuscular, and sensory systems. Usually, children initially walk independently at 12 months of age, although the time of onset may be variable between 8 and 18 months [48]. Walking consists of a process of moving in a definite direction with a series of continuous and coordinated movements of the hip, knee, foot, and ankle joints, during which, in addition to the coordination of movements, the support of the body in space is necessary to perform the movement [48]. For example, factors in body size, stability of head and trunk posture, muscle/fat ratio, leg length, ossification of bones, and appearance of arch structures are important for the mastery of walking [48,49]. Children’s motor development does not have a strictly defined time frame and is nonlinear, given the many different interactions and contexts, in other words, gait changes not only occur at different times for each infant but also with the use of different strategies [3,48,49,50].

Studies on the biomechanical characteristics of gait at the beginning of independent walking in typically developing infants have divided gait maturation into two stages [48,51]. During the first stage, toddlers exhibit relatively high gait frequency, a broad base of support, extended stance time, and increased foot support, which are elicited by moving the entire body in steps to regain balance without initially pushing away from the point of contact with the ground. Active heel dorsiflexion does not occur until 2 years of age. Dynamic balance and trunk stability during walking are improved three to six months after the beginning of independent walking. The kinetic features of immature gait in toddlers are the predominance of hip and knee extension moments in the entire stance along with the sustained development of force observed around these joints [24,51,52]. With the accumulation of experience in the period between 1 and 4 years of age, owing to neuromuscular maturation and changes in foot morphology, young children gradually increase normalized stride length and speed and decrease stride width and gait frequency, i.e., narrow the base of support. The transition from early contact with flatfoot to heel strike usually occurs in toddlers 12 months after the start of independent walking as a transition to adult gait [48]. Many toddlers can walk on their toes when weight transfer occurs only in the forefoot and not from heel to toe during the transition to adult gait. There is no or limited heel strike in the contact phase of the walking cycle and no dorsiflexion of the foot during the stance phase [19].

The second stage (up to 7 years of age) is characterized by a gradual transition to a fine-tuned gait [48]. Developing the ability to use plantar flexion movements is completed by 4–5 years of age [24]. As toddlers develop, their walking style becomes more refined. A mature gait is characterized by the heel making contact with the ground first, followed by forward movement under the ankle in the middle position, and finally concluding with a forefoot push-off. This is a sign of proper development and coordination in young children. The distribution of foot load during heel contact in young children mainly arises in the anterior human foot, which undergoes a significant transformation during growth. For instance, the calcaneus, or heel bone, and the narrow heel of a baby’s foot can handle loads transmitted in the anterior-posterior and medial directions up to the age of six years [48]. This means that as a child’s foot develops, it becomes better equipped to support their weight and coordinate their movements. Becoming more upright in the trunk, the generation of the so-called superimposed tissues of the body parts, and a gradual pattern of mutual activation between the lower limb antagonists instead of co-activation [24] will, in turn, come at this time.

## 3. Brain Development

The divisions of the central nervous system (CNS) providing the motor control necessary to maintain bipedal posture and locomotion develop earlier than the peripheral musculoskeletal components essential for walking, preceding the functional emergence of these skills in infants [53].

Fetal brain development begins in the first trimester of pregnancy and continues postnatally. The fetal brain neocortex is built from 5 to 20 weeks of gestation [54]. Migration of crucial neurons from the ventricular zone to the neocortex occurs at the end of the first and during the second trimester of pregnancy [55]. It is necessary to control low concentrations of the cytokines IL-6 and TNF-α to regulate the proliferation, differentiation, and survival of neuronal cells in the fetal brain. Physiological levels of IL-6 regulate the normal formation of dendritic spikes, but high levels of IL-6 could lead to abnormal patterns of spike formation, thereby disturbing synaptogenesis and the equilibrium between excitatory and inhibitory signal transduction in the limbic system. TNF-α participates in the regulation of synaptic plasticity and the enhancement of synaptic transmission induced by astrocytes through stimulation of glutamate release, whose elevated levels in inflammation can induce glutamate-mediated cytotoxicity. According to epidemiological studies, the first trimester of pregnancy is the most vulnerable period when offspring are most likely to develop neuropsychiatric behavioral phenotypes. Infection during pregnancy can lead to negative impacts on the fetal CNS. This is due to maternal immune activation (MIA), which can affect the fetus through the placenta, amniotic fluid, and maternal serum. The criticality of gestational age at the time of exposure to MIA for further behavioral changes was confirmed in models of MIA induced by lipopolysaccharides (LPS) of the Gram-negative bacteria cell wall. This may be due to the fact that dopaminergic motor behavioral changes continue into adulthood. Although prenatal immune provocation can prime CNS disease, genetic and environmental factors also determine the probability of its development [55].

Elevated levels of the cytokines IL-6 and TNF-α, along with such proinflammatory cytokines as IL-17a, IL-1β were also observed in ASD [55]. Short-term local inflammatory reactions are considered not to affect hippocampal neurogenesis, although persistent inflammation can affect abnormal cortical development [55,56]. Chronic overexpression of the proinflammatory cytokine IL-1β in the hippocampus inhibited hippocampal neurogenesis in adult humans, although the anti-inflammatory cytokine IL-10 also reduced neuronal differentiation and neurogenesis in the adult rodent brain [56]. However, their function during postnatal brain development remains a gap.

The functional networks of the human brain, especially the primary visual, auditory, and sensorimotor networks, and the default-mode and executive-control networks involved in heterogeneous functions are topologically structured. During the third trimester of pregnancy, the brains of full-term babies experience rapid growth and interconnection of neurons, which contributes to the development of primary sensorimotor skills and higher cognitive functions. During this period, the functional network of the brain undergoes tremendous remodeling, resulting in a higher degree of brain organization, which is why preterm birth is associated with poor neurodevelopmental outcomes [57]. Moreover, preterm infants have a high probability of evolving motor development delays [58] and severe gut microbiota dysbiosis [57].

The neural structures for visual–motor interactions are present at birth. A well-defined primary sensorimotor cortex has been detected in the neocortex of the newborn on postnatal day six [54]. The brain circuits of the neocortex, formed by a great variety of excitatory and inhibitory neuronal subtypes located in its layers and columns [54], are plastic, and motor and sensory learning might shift them at many levels with stable synaptic changes appearing in their structure depending on acquired experience [59]. The development of the cellular and synaptic organization of the neocortex brings about not only a set of sensory and motor functions but is also accompanied by the development of cognitive abilities [54]. The dominant development of primary regions prior to birth, supported by a dramatic increase in short- and mid-range connections between them, is suggested to be beneficial for basic survival functions by the time of birth. In contrast, the development of long-distance connections, primarily related to global information integration, mainly occurs after birth [57]. The brain keeps developing intensively after birth, increasing in volume from about 36% to 80–90% of its adult volume by the age of 2 years [60]. Significant modifications in the relationship between functional brain connectivity and the development of their walking ability and general motor skills were demonstrated in infants and toddlers at ages 12 and 24 months. At 12 months of age, a remarkable impact on the development of walking abilities was observed. The discovery has revealed a profound connection between the functional connectivity of the motor and default mode networks, providing valuable insight into the intricate relationship between these two essential components. When children reach the age of 24 months, their development process reveals the emergence of additional networks, including the dorsal attention and posterior cingulo-opercular networks. These networks played a significant role in enhancing their walking abilities, further supporting their overall growth. The assessment of overall gross motor function indicated the participation of motor and default mode networks at 12 and 24 months. Moreover, at the age of 24 months, the dorsal attention, cingulo-opercular, frontoparietal, and subcortical networks were also found to be involved [61].

## 4. Visual–Motor Learning and Brain Control of Movement

In a study by Donne et al. [31], it was shown that there was a difference in activation of brain regions associated with somatosensory discrimination (supramarginal gyrus) and with visuospatial processing and episodic memory (collateral gyrus), responsible for sensory decision-making, in children with and without ITW. This change in activation may reflect the impairment of visual–motor learning and the lack of error correction during this learning in children with ITW.

The infant has to control the whole body at the same time as performing the movement to maintain its balance in space (postural control). This movement requires that infants learn to generate and use perceptual information about the current state of their bodies concerning their environment, obtained through visual and tactile feedback [52]. Until three years old, vision plays a dominant role in motor learning [50] and, in particular, in postural control. Toddlers may have problems with postural control when performing several tasks [3,50], sometimes making movements that could be interpreted as preemptive corrections when performing complex play tasks.

Around age 4 to 5 years, there is a pivotal moment in which children gain the ability to integrate information from multiple sources, leading to a significant advancement in their anticipatory control and integrative processes. However, when it comes to choosing walking speed and stride length, they still mostly rely on visual information. From three to six years old, kids acquire the skill of coordinating their head and torso movements while walking, leading to better management of their legs. During this period, they learn to resolve inter-sensory conflict. Once a child reaches the age of 7, they acquire the capability of selecting the most pertinent information source required for maintaining their posture and will not be as reliant on visual cues [3,50].

Visual acuity, i.e., the ability to perceive small details, develops rapidly during the first months of life and persists until 3–4 years of age, but it is significantly related to general motor function and not to fine or gross motor function as reported in children at age 4.5 [62,63,64,65].

The sensorimotor cortex, premotor cortex, basal ganglia, and brainstem structures are involved in walking initiation. The basal ganglia, premotor cortex, supplementary motor area, and motor cortex participate in the generation of anticipatory postural correction [66].

Before performing a directed movement to achieve a visual goal, the brain has to focus on a given task and generate a motor plan for the movement. The brain performs permanent movement control and makes corrections using a forward model predicting sensory motor command response consequences, which are mounted on an action understanding based on experience [64]. The motor cortex and related brain regions generate motor plans and translational movement patterns. The generation occurs in areas of the temporal lobe, such as the fusiform gyrus; areas of the parietal lobe, such as the dorsomedial posterior parietal cortex, the caudal part of the dorsal premotor cortex, the superior parietal lobule, and the supramarginal gyrus; the anterior and posterior compartments of the intraparietal sulcus; and frontal areas, such as the inferior frontal gyrus [2].

As an error control system, striatal dopamine neurons perform the movement error correction of a mismatch between the actual result of the motor command and the predicted sensory consequences (Figure 2).

A decrease in dopamine levels from the expected reward level in the case of successful prediction indicates an error that results in a new movement plan and forward movement pattern. The correcting motor error process activates the observation–execution system, known as the mirror neuron system, consisting of the inferior frontal, premotor, and inferior parietal brain areas. While the child develops, body dynamics and experience constantly change, allowing the brain to acquire and renovate movement patterns.

A study of the development of kinematic coordination patterns in the movement of toddlers suggests a constant renewal of the neural command when changing morphological variables as the child grows [2,67]. Despite intensive studies of motor planning issues, including in children [68], it is not yet clear how it is carried out in children when learning to walk.

## 5. Dopamine Signaling

The dopaminergic pathway is formed by dopamine-producing neurons located in the ventral region of the cap and the compact part of the middle brain substantia nigra projecting to the adjoining nucleus via the mesocortical or mesolimbic pathways and the dorsal striatum via the nigrostriatal pathway, respectively. Both the mesocorticolimbic and nigrostriatal pathways are associated with motor control, reward, and habit learning (Figure 2). The dopamine-producing neurons of the mesocorticolimbic pathway are responsible for movement planning and correction [69,70], and possibly for the formation of gait patterns. The neurons of the nigrostriatal pathways regulate tone and contraction in skeletal muscles to a greater extent [70,71].

The presynaptic membrane at the dopaminergic endings locates the dopamine transporter (DAT), which regulates intracellular and extracellular dopamine. Five types of dopamine receptors (D1–D5) [72] grouped into the D1- and D2-like receptor families regulate numerous dopamine functions, including cognition, reward, and voluntary motor movements [73]. Of note, the expression of DAT and D1- and D2-like receptors also seems to be regulated by sex hormones, which could affect the difference in locomotor function manifestation, including in neuropsychiatric disorders [74,75]. Cesarean section, as shown in rats, is also related to factors increasing D1 receptor expression in areas of the striatum and hippocampus [60].

Activation of D1-like receptors often increases neuronal excitability, while activation of D2-like receptors usually decreases neuronal excitability [73]. A recent study reveals a direct role of the D1 receptor in the modulation of its downstream-mediated control of visual cortical activity in non-human primates [75].

The possibility of controlling movement by modulating the tone and contraction in skeletal muscles through the concentration-dependent action of nigrostriatal dopamine on postsynaptic D1- and D2-receptors was also discussed. During movement, the concentration of dopamine increases sequentially from nanomolar levels interacting with the D1 receptor, which inhibits muscle tone, to micromolar levels interacting with the D2 receptor, thereby accelerating muscle contraction [71].

Dopamine neurons are referred to as one of the most sensitive types of neurons to both intrinsic and extrinsic stressors in the brain [76]. Intestinal microbiota has been described among the extrinsic ones. It can modulate the dopaminergic pathway both positively and negatively by altering dopamine production through the metabolism of its precursors, affecting the expression of dopamine receptors D1 and D2 and the DAT transporter on specific dopamine targets, and altering brain neurochemistry through vagus-mediated pathways and the metabolites they produce. For example, different mechanisms underlying the D1 and D2 receptors are controlled by the gut microbiota of various genera: *Prevotella*, *Bacteroides*, *Lactobacillus*, *Bifidobacterium*, *Clostridium*, *Enterococcus*, and *Ruminococcus* [72]. The administration of *Bacteroides uniformis* via fecal microbiota transplantation to mice promoted an increased ability to bind DAT to dopamine in the striatum, whereas *Prevotella copri* had the opposite effect [72]. Notably, in a study of the Danish population, *P. copri* was the most abundant bacterium in the gut microbiota of 6–9-year-olds but not in adults [77]. Once intestinal dysbiosis affects these microorganisms, dopaminergic deficiency can develop [72].

The metabolic abnormalities developing in gut dysbiosis vary in their manifestations. For example, the dopamine and serotonin (5-hydroxytryptamine, 5-HT) metabolism-associated pathways are mostly altered in ADHD [78]. Dopaminergic stimulants administered to children with ADHD improve motor functions such as dynamic balance, coordination, and fine and gross motor skills [2]. However, there may be the opposite effect, as in the case of inhibition of dopamine-β-hydroxylase by metabolites produced by *Clostridia* from glyphosate, when the conversion of dopamine to norepinephrine is blocked, and its excess, which has a toxic effect on cells, occurs [79]. Despite levels of glyphosate being widely accepted in agriculture and considered safe for health [80], there is evidence of the development of ASD-like conditions in children due to the chronic presence of glyphosate in the diet [79,81]. Drinking water containing the glyphosate-based herbicide during pregnancy and lactation in rats impaired sensorimotor function in pups [82]. The developing dopamine system is sensitive to early changes and disturbances in the gut microbiota [57], since this often provokes inflammation. LPS-induced inflammation potentially increases dopamine release in the short term, whereas chronic exposure conversely decreases dopamine release in the striatal body and D2 receptor binding [72].

## 6. Microbiome and Motor Development

### 6.1. Colonization of the Neonatal Intestine and Development of Gut Microbiota in Children

The gut microbiota is a dynamic and complex community of microorganisms that resides in the human gastrointestinal (GI) tract. Dominated by bacteria, this community is also home to archaea, eukaryotes, and viruses. Their colonization of the gut in newborns is an orderly process that goes hand in hand with the development of the child’s various body systems. The gut microbiota has a beneficial effect on the host, influencing host development, physiology, and metabolism [83]. Numerous articles and reviews have thoroughly explored and outlined the assembly of gut microbiota, along with the effect of diverse factors on this process [77,84,85,86,87]. Research shows that the initial three years of a child’s life play a critical role in the establishment of their gut microbiota. However, the development of this microbiota towards adulthood undergoes a continuous process for several more years. During this period, the gut microbiota is not yet sufficiently stable and can be modified [77,84,85,86,87]. The colonization of the infant’s gut by bacteria is normally triggered by the microbiota of the birth canal, colostrum, and breast milk. *Bifidobacterium* spp. and *Bacteroides* spp. transmitted to newborns from mothers share the same enzymes for mucin and human milk oligosaccharide degradation, which contributes to their selective support in the gut [88]. *Bifidobacterium* and *Bacteroidetes* are keystone taxa in maintaining and stabilizing the gut microbiota, especially since *Bacteroides* spp. exhibit functional redundancy [89,90]. The components of breast milk, in particular oligosaccharides and glycans of glycoproteins, and immunoglobulins, which vary throughout lactation, act as a vector that guides the formation of the composition and structure of the initial microbiota of the first year [91]. The glycan composition of breast milk in different mothers is determined by the activity of the *FUT2* and *FUT3* genes encoding α1,2-fucosyltrasferase and α1,3-fucosyltrasferase, respectively. Fucosylated glycans contribute to the predominance of *Bifidobacterium* spp. and the retention of small *Bacteroides* spp. at this stage [85,89]. In the first year of life, *Lactobacillus* spp. are the most numerous intestinal bacteria after *Bifidobacterium* spp. [92]. The introduction of complementary foods leads to an increase in the alpha diversity of the microbiota, in particular representatives of the phylum *Bacteroidetes*, *Firmicutes*, *Proteobacteria*, and *Verrucomicrobia*. Representatives of different bacterial taxa show different tendencies to increase in abundance and diversity during this period [77,84,86,87]. In the period from 9 to 18 months, there is an increase in the relative abundance of representatives of the families *Lachnospiraceae*, *Ruminococcaceae*, *Eubacteriaceae*, *Rikenellaceae*, or *Sutterellaceae*, and a decline in representatives of *Bifidobacteriaceae*, *Actinomycetaceae*, *Veillonellaceae*, *Enterobacteriaceae*, *Lactobacillaceae*, *Enterococcaceae*, *Clostridiales incertae sedis* XI, *Carnobacteriaceae*, and *Fusobacteriaceae*. Between 15 and 30 months, only *Bacteroidetes* and *Proteobacteria* continue to develop, after which their presence and alpha diversity stabilize [77,87].

### 6.2. Control of Brain Development by Microbiota

The age of about two or three years is one of the critical windows in a child’s development [93,94]. During this period, active synaptogenesis, glia expansion, and myelination occur that are closely related to the development of the child’s gut microbiota, changing its composition and structure towards being closer to the more stable and complex adult gut microbiota at this time as well [60,94]. The normal gut microbiota mediate the expression levels of two synaptic proteins, synaptophysin and psd-95, thus modulating synaptic development. In addition, it affects the myelination of axons in the prefrontal cortex [60]. Microglia, or resident macrophages of the CNS, have a major role in both the developing and mature nervous systems. In addition to its immune function, it is involved in the regulation of synaptic transmission, synapse contraction and formation, cell death and survival, embryonic wiring, and the formation of a functional syncytium, ensuring the stability of the nervous microenvironment, especially under metabolic stress [95]. Fetal brain microglia development is controlled by the maternal microbiota during pregnancy. Studies on rodents have demonstrated developmental stage- and sex-dependent regulation of fetal microglia gene expression by the maternal microbiota through modulation of chromatin availability; its absence during this period critically affects microglia development. At the same time, the basic signature of microglia gene expression is highly conserved in humans and mice. Sexual dimorphism of transcriptomic profiles in microglia cells appears postnatally [95]. Postnatal neurogenesis in the hippocampus in mice is also regulated by the intestinal microbiota and correlates with age-related changes, the appearance time of which is determined by sex [96]. The peculiarities of neurogenesis associated with sex and microbiota probably explain the prevalence of such neuropsychiatric and neurological disorders as ASD, ADHD, early onset schizophrenia, and Parkinson’s disease (PD) [97,98].

Experiments using germ-free (GF) mice and normal mice indicated that colonization with intestinal microbiota at an early age was necessary for normal development of the hypothalamic-pituitary-adrenal axis and neuroendocrine stress response, as well as for expression of brain-derived neurotrophic factor (BDNF) by the cortex and hippocampus [99]. BDNF, as a protein involved in the regulation of synaptic plasticity and neurogenesis in the brain, is important for motor learning and memory formation [100]. Its expression in the motor cortex was highest in juvenile mice when adult motor patterns were shaped, while BDNF levels in adults were low [100,101]. BDNF protects dopamine neurons from degeneration [72]. A decrease in BDNF expression in the hippocampus was observed only in male GF mice, which is associated with the dependence of the serotonergic system of the CNS on estrous cycle hormones [102]. This implies that disorders of microbiota maturation at an early age can result not only in impaired brain development but also in an imbalance in neurotransmitter signal transmission, as well as greater sensitivity to stress during this period in males [99,103]. The presence of the genera *Bifidobacterium* and *Lactobacillus* in the microbiota correlated most positively, while the presence of representatives of the families *Bacillaceae*, *Enterococcaceae*, *Erysipelotrichaceae*, *Lachnospiraceae*, *Paenibacillaceae*, and *Veillonellaceae* showed a negative correlation with BDNF mRNA expression. For members of the families *Ruminococcaceae*/*Oscillospiraceae*, *Turicibacteraceae*, *Eubacteriales* Family XIII, and *Clostridiaceae*, both positive and negative correlations with BDNF mRNA expression have been reported [103].

### 6.3. Impact Factors on Colonization of the Neonatal Intestine

Colonization of the neonatal intestine depends on a variety of endogenous and exogenous factors, the most critical of which are mode of delivery, gestational age, feeding practices, and the use of antibiotics during pregnancy [93,104]. Normal infants obtain bacteria such as *Bifidobacterium* spp. and *Bacteroides* spp. from the mother during breastfeeding and vaginal delivery; a lack of breastfeeding prolongs a delay in the colonization of the intestine by these bacteria [105]. Over the past decades, infant breastfeeding and its duration, especially after cesarean section, have declined dramatically worldwide [106]. At the same time, the global use of cesarean sections is increasing; by 2018, its use reached an average of 21% with an annual increase of 4% [107]. The level of *Lactobacillus* genus remained significantly lower in cesarean section infants during the first 6 months of life compared to vaginally delivered infants [87]. The effect of cesarean section on gut microbiota is normalized in 3–5-year-old children [84]. It has been reported that active colonization by opportunistic pathogens, especially *Enterococcus faecalis*, is observed in cesarean sections, and the lack of several *Bacteroides* species in newborns is a distinguishing feature of such deliveries from vaginal deliveries [108]. Referring back to the already described MIA, it is worth noting that treatment of *Bacteroides fragilis* mouse models of ASD contributed to a decrease in IL-6 levels and, along with other symptoms of ASD, improved sensorimotor disorders [109]. It has also been reported that cesarean sections delay development in the gross motor domain at 9 months of age but not at 3 years, along with developmental delays in the social domain [60]. In addition, *Bacteroides thetaiotaomicron* regulates the afferent innervation of neurons and the vagus nerve, as has been proven [110]. Moreover, various pesticides and herbicides, particularly glyphosate, which are widespread in agriculture, have also recently become a frequent cause of intestinal microbiota dysbiosis [81,111]. The glyphosate target is the shikimate pathway, which is present not only in plants but also in several bacteria. Consequently, it reduces the number of bacteria that do not have enzymes for their degradation, increasing the level of resistant bacteria [112], which provokes a metabolic shift in the intestinal microbiota metabolome towards an increase in propionate production [99]. Antibiotics have led to an imbalance in the gut bacteria, which resulted in higher levels of L-DOPA in the prefrontal cortex and hippocampus. At the same time, there was a decrease in hippocampal 5-HT levels and a lowering of the homovanillic acid (HVA)/dopamine ratio in the amygdala and striatum in rats compared to the control group in rats [113]. In addition, the results of the symptomatic preterm Neonates (REASON) study have demonstrated that antibiotic use in preterm infants leads to dysbiosis, in particular a decrease in the level of lactate-metabolizing *Veillonella. Veillonella* may be involved in the biosynthesis and export of L-glutamate [114]. In general, dysbiosis of the gut microbiota possibly increases gut and blood–brain barrier (BBB) permeability by contributing to the entry to the bloodstream and CNS of gastrointestinal contents, including bacteria and LPS, leading to systemic inflammation of the peripheral and central nervous systems and possibly, eventually, to neurological disease [99].

Prenatal stress can alter fetal brain development, provoking reprogramming of the hypothalamic–pituitary–adrenal axis, cognitive deficits, and behavioral abnormalities later in life. The intestinal microbiome of infants whose mothers suffered prenatal stress was characterized by a lower content of lactic acid bacteria and bifidobacteria [60].

Preterm birth is the highest risk factor for ITW in babies [34,35], along with induced labor [34]. Mothers who give birth prematurely have been found to have a unique variation of *Lactobacillus* present in their vaginal microbiomes, according to a recent study [115]. This sets them apart from mothers who carry their babies to full term. In premature births, the levels of *Lactobacillus* were lowered while the presence of *Gardnerella* and *Prevotella* increased, leading to the promotion of proinflammatory cytokines. Furthermore, these alterations in the vaginal composition can potentially trigger intestinal dysbiosis in newborns.

The neurochemical effects of gut microbiota dysbiosis on the hippocampus at an early age are extremely hard to correct later in life [102], but early intervention for their prevention using probiotics to restore a normal balance of microorganisms in the human gut seems promising. A randomized clinical trial suggested that administering *Lactobacillus rhamnosus* GG as a probiotic to infants starting at 6 months of age helped reduce the risk of ADHD in later childhood [116].

### 6.4. Link between Infant Nervous System Development and Gut Microbiota Content

Recent research explicitly linking the development of the infant nervous system to the development of the gut microbiota has begun to emerge. In a prospective longitudinal study, general cognitive activity and the development of specific functional brain areas in infants and toddlers at the ages of one and two years were assessed for parameters such as gross and fine motor skills, visual perception, receptive and expressive speech, and their gut microbiome [117]. Analysis of metagenomic data of infants’ gut microbiomes revealed the presence of three clusters distinguished by a relatively high abundance of key bacterial genera, including *Faecalibacterium*, *Bacteroides*, and an unnamed genus in the family *Ruminococcaceae*. Infants with bacterial cluster 2 scored highest on the assessed parameters, especially compared to infants with bacterial cluster 1. They were more frequently breastfed and were vaginally delivered compared to infants with other bacterial clusters [117]. Another CHILD (Canadian Healthy Infant Longitudinal Development) cohort study assessing neurodevelopment by the Bayley Scale of Infant Development (BSID-III) also identified three groups of infants based on the relative abundance of the gut microbiota at 12 months of age. In this study, a cluster dominated by *Bacteroides* was also associated with higher scores in cognitive function, language, and motor skills, but only in male infants [118]. A study exploring the connection between intestinal microbiota and fine motor skills in 18-month-old full-term healthy infants focused on two enterotypes: the first based on *Bacteroides* and the second based on *Firmicutes*. However, in this study, children with a *Bacteroides*-dominant community showed lower fine motility scores, while better scores were related to an increase in eleven genera: *Collinsella*, *Coprococcus*, *Enterococcus*, *Fusobacterium*, *Holdemanella*, *Propionibacterium*, *Roseburia*, *Veillonella*, an unidentified genus within *Veillonellaceae*, and especially *Lactobacillus* and *Bifidobacterium*. In this study, the relationship of indices with sex was not observed [118]. All these results illustrated the importance of the intestinal microbiota not only for cognitive development but also for the development of the child’s motor skills. The contribution of *Bacteroides* spp. to neurogenesis in children is not entirely clear, although perhaps through the metabolites they produce, including probably sphingolipids and ceramides [119,120,121,122]. Conflicting data on *Bacteroides* spp. have also been reported in the case of ADHD in children. For example, impaired cognitive function correlates with both reduced contents of *Bacteroides ovatus* [123] and increased contents of *Bacteroides uniformis* and *B. ovatus* [124]. Probably, depending on the changes in their composition and level, different manifestations of ADHD can be observed [123,125], since a direct correlation with ADHD symptoms has been evidenced for *B. ovatus* and *Sutterella stercoricanis* [125]. A similar effect of the presence of certain representatives of the intestinal microbiota could be expected in ITW.

### 6.5. Microbiota–Gut–Brain Axis

The existence of brain-mediated control of the intestinal microbiota composition has been demonstrated in humans with brain injury [126] and in a mouse model of CNS damage in which a marked decrease in *Lactobacillus gasseri* abundance was observed [127].

The microbiota, gut, and brain interact through a dynamic bidirectional microbiota–gut–brain axis. The main interaction mechanisms within the axis include the immune system, the vagus nerve (*nervus vagus*), the enteric nervous system (ENS), and microbial metabolites such as neurotransmitter (NT) precursors, peptidoglycans, and short-chain fatty acids (SCFAs) [128,129,130] that are absorbed into the bloodstream and can interact with enzymes and receptors expressed by the host. There are two main pathways of the “microbiota–gut–brain” axis at the neuroanatomical level: the direct pathway between the gastrointestinal tract and the brain represented by the vagus nerve and the autonomic nervous system (ANS), and the indirect pathway carried out jointly by the ANS and the ENS [130].

Several studies have demonstrated the ability of some probiotic bacterial strains to modulate NT functioning in humans and rodents, as discussed in reviews [56,99,128,131]. For example, *L. rhamnosus* increases the excitation frequency of vagus nerve afferent impulses, activating dopaminergic neurons and increasing the concentration of dopamine in the brain [131]. In addition, when administering *L. rhamnosus*, the expression of GABA1β receptor mRNA enhances in the cortex and decreases in the hippocampus, amygdala, and locus coeruleus, while the expression of GABAα2 mRNA increases in the hippocampus and decreases in the prefrontal cortex and amygdala [128]. Heat-killed *Lactobacillus brevis* enhanced the survival of hippocampal neurons [56]. *Lactobacillus plantarum* increased the expression of BDNF, 5-HT, 5-HT transporter, and neurotrophin in the brain while also increasing the concentration of 5-HT in the gut. *Bifidobacterium breve* contributed to the increase in BDNF levels in the hippocampus and suppression of hippocampal inflammation, and combined administration of *Lactobacillus helveticus* and *Bifidobacterium longum* attenuated stress-induced suppression of hippocampal neurogenesis [56]. *Lactobacillus reuteri* and *Bifidobacterium Teenis* administration improved BDNF production in the hippocampus of mice with anxiety/depression and colitis [122]. At the same time, some bacteria can not only increase but also decrease NT levels. For example, *Enterococcus faecium* and *E. faecalis* can convert tyrosine or L-3,4-dihydroxyphenylalanine (L-DOPA) into dopamine, but *E. faecalis* can also deplete dopamine precursors in the GI tract [72]. Also, *E. faecalis* promoted the restoration of neurogenesis in the hippocampus, which was inhibited by inflammatory bowel disease [56]. Several *Clostridium* species can be classified as neuropathogenic since they degrade dopamine, as *Clostridium tetani* does [72].

Administration of probiotics such as *B. longum* 1714 to healthy adult volunteers, *Lactobacillus acidophilus* (strain Rosell-11), and a mixture of *L. acidophilus*, *L. rhamnosus*, and *B. longum* to children with ASD (aged 4–10 years and 5–9 years, respectively) showed a positive effect on brain health, which has led them to be recently often considered psychobiotic [132,133]. Because probiotics can potentially restore the normal balance of microorganisms in the human gut and normalize the processes of neurogenesis, myelination, microglia cell maturation, and preservation of the integrity of the BBB, they are being explored as a promising therapeutic supplement in the treatment of various neurodegenerative diseases [104].

## 7. Metabolites of the Intestinal Microbiota

The normally developing gut microbiota provides the brain with various metabolites that influence the long-term programming of the neocortex brain circuits, including its involvement in the regulation of motor control and cognition [60]. The intestinal microbiome and the levels of its produced metabolites dynamically change during the first 18 months of life [134].

### 7.1. Neurotransmitters

NTs are low-molecular-weight compounds that participate in the transmission of information through the CNS and peripheral nervous system, as well as in the regulation of neuronal growth, differentiation, and survival [128]. NTs such as glutamate, γ-aminobutyric acid (GABA), dopamine, and 5-HT are involved in the neuroregulation of processes related to motility [135]. NTs are of both endogenous and exogenous origin. A wide spectrum of bacteria is capable of producing NTs and their precursors [136,137]. Sterile mice have been demonstrated to have lower circulating concentrations of NTs such as 5-HT, dopamine, and GABA [99]. Although glutamate, GABA, dopamine, and 5-HT produced by the gut microbiota cannot penetrate through the BBB and reach the brain, they can affect the brain via modulation of the immune system, altering cytokine production by immune cells [99]. NT precursors, such as acetate (a precursor of glutamate and GABA), entering the bloodstream from the intestine or being injected intravenously, can be incorporated into the metabolic pathways of NT synthesis in the brain. Tyrosine and tryptophan derived from diet can also enter from the intestine into the bloodstream, cross the BBB, and be absorbed by NT-producing cells in the brain [137]. The latter means potential competition for them between the host and the bacterial NT-producing cells.

Glutamate acts as the predominant excitatory NT in the CNS. It can be derived from glutamine and is a precursor of GABA. Its release from presynaptic neurons into the synaptic cleft activates n-methyl-d-aspartate (NMDA) and α-amino-3-hydroxy-5-methyl-4-isoxazolepropionic acid (AMPA) receptors. Glutamate provides calcium and sodium influx to postsynaptic neurons, and its excess might cause extreme neuronal excitation and excitotoxicity. Glutamatergic neurotransmission is involved in long-term potentiation, contributing to cognitive functions such as learning and memory formation as well as motor, sensory, and autonomic CNS functions. Excessive glutamate content is potentially associated with neurological conditions such as multiple sclerosis (MS), amyotrophic lateral sclerosis, and PD [128].

GABA is produced by interneurons but also has a microbial origin. It is the major inhibitory NT in the adult brain, but within the short postnatal period, GABA is an excitatory NT, triggering depolarization instead of hyperpolarization in various areas of the nervous system (e.g., neocortex, hippocampus, hypothalamus, cerebellum, and spinal cord). Its low levels are supposed to be responsible for the increased excitability of neurons. Maintaining a balance between inhibitory and excitatory transmission is essential for normal brain function. Altered GABAergic neurotransmission can also cause neurological disorders (ASD, schizophrenia, and epilepsy) and neurodegenerative diseases (e.g., MS, Alzheimer’s disease, and PD).

GABA and glutamate levels are modified in children with ASD, creating an imbalance between excitatory and inhibitory mechanisms [128].

Dopamine appears in almost all centrally controlled events, from motor control to cognitive functions [128].

5-HT, another important NT, is involved in the regulation of sleep and wakefulness states, gastrointestinal secretion and peristalsis, respiration, vasoconstriction, behavior, and neurological function. Almost 95% of the body’s 5-HT is produced by enterochromaffin cells in the gut, which can increase its production under the influence of SCFA produced by the intestinal microbiota. The remaining 5% is mainly produced by rostral and caudal groups of neurons in the suture nuclei in the brainstem that further project into the cerebral cortex, thalamus, hypothalamus, and basal ganglia, as well as into the brainstem and spinal cord, respectively [128,138]. Besides, the source of 5-HT is the metabolism of tryptophan by the intestinal microbiota through the kynurenine pathway [138]. Research has shown that the depletion of tryptophan could lead to a decline in verbal information processing among individuals who suffer from PD. This finding suggests a potential relationship between the impairment of the serotonin and acetylcholine systems. In contrast, young individuals who suffer from ADHD do not exhibit impaired memory function [138].

Tryptophan in the gut might be metabolized by the microbiota in three ways: into 5-HT, kynurenine, and indole. Less than 5% of tryptophan is converted into 5-HT, and the remaining 95% is metabolized mainly through the kynurenine pathway by the enzymes tryptophan-2,3-dioxygenase (TDO) and indoleamine-2,3-dioxygenase (IDO), while the tryptophanase enzyme plays a role in converting tryptophan into indole. Indole acts as a precursor for several neuroactive signaling molecules and also stimulates enteroendocrine L cells to secrete glucagon-like peptide-1 (GLP-1) and regulate intestinal barrier permeability. LPS-initiated inflammation induces IDO expression and shifts the kynurenine/tryptophan ratio upward. Increased activity of the kynurenine pathway can produce the neurotoxic quinolinic acid metabolite and decrease central serotonergic availability [99].

5-HT inhibits dopamine release, modulates glutamate and GABA transmission, inhibits glutamate release in the frontal cortex, and enhances glutamate transmission in the prefrontal cortex [99].

Motor dysfunction often represents the very first sign of an imbalance in the functional balance of neurotransmitters produced in the brain [139].

### 7.2. Group B Vitamins

The intestinal microbiota produces a crucial set of vitamins known as group B vitamins. These vitamins play an essential role in several metabolic processes, including amino acid, glucose, and fatty acid metabolism. As such, they are vital to maintaining good health and wellbeing. Moreover, they aid in the synthesis and metabolism of neurotransmitters and neurohormones, like 5-HT, dopamine, adrenaline, acetylcholine, GABA, glutamate, d-serine, glycine, histamine, and melatonin. Additionally, they are involved in tryptophan metabolism by the kynurenine pathway and homocysteine metabolism. This shows the significance of having a healthy gut microbiome in maintaining overall health, especially concerning brain function and metabolism. Metagenomic analysis of the human gut microbiota indicated 49% of the metabolic pathways associated with vitamins in the phylum *Firmicutes*, 19% in *Proteobacteria*, 14% in *Bacteroidetes*, and 13% in *Actinobacteria*. There are complex interactions between vitamin-producing bacteria and the bacteria that consume them. Gut dysbiosis can lead to vitamin deficiency, which in turn reduces the synthesis of vitamin-dependent neurotransmitters, causing an imbalance that can result in central nervous system dysfunction [140].

Pyridoxal phosphate (Pyridoxal-5’-phosphate), which is the active form of vitamin B6, is a cofactor of enzymes involved in the metabolic pathways of 5-HT, dopamine, GABA, glutamate, d-serine, glycine, histamine, and melatonin synthesis. Vitamin B2 (riboflavin) is necessary to convert pyridoxine and folic acid into bioavailable forms. Pantothenic acid, also known as vitamin B5, is a precursor of coenzyme A (CoA) needed for myelination and acetylcholine synthesis. Vitamins B2 and B6 are enzymatic cofactors at various stages of the kynurenine pathway. Flavin adenine dinucleotide (FAD), an active (coenzyme) form of riboflavin, is involved in tryptophan and kynurenine metabolism. That makes vitamin B2 an important regulator of niacin synthesis and niacin-containing coenzymes such as nicotinamide adenine dinucleotide (NAD) and nicotinamide adenine dinucleotide phosphate (NADPH). There are various mechanisms related to immune inflammation that can trigger the kynurenine pathway, resulting in a reduced availability of tryptophan for the synthesis of 5-HT and melatonin. Deficiency or changes in the concentration of these vitamins can accumulate or enhance the metabolism of various kynurenines and disrupt the conversion of tryptophan and kynurenines to NAD and nicotinic acid (NADH), also known as vitamin B3, thus possibly playing a crucial role in the pathogenesis of psychiatric and neurological disorders [140]. Folic acid, also known as vitamin B9, can potentially increase the levels of 5-HT and improve the expression of BDNF and glutamate receptor 1 (GluR1) in the brains of rats with depression. This can be observed in both the hippocampus and association cortex, indicating a positive effect on both memory and cognitive abilities [99]. It is noteworthy that the microbiota of kids harbors more genes that produce vitamin B9, unlike adults, who primarily rely on dietary sources to obtain this vitamin. Although higher levels of vitamin B12 are typically found in adults, it is crucial to underscore the critical role this nutrient plays in enhancing neurological function [141] and motor skill development [142], particularly in children around 7 years old. This once again highlights the importance of maintaining a healthy gut microbiome to ensure adequate nutrient production and absorption throughout all stages of life.

### 7.3. SCFAs

Commensal microorganisms can produce SCFAs such as acetate, butyrate, and propionate derived from the fermentation of breast milk oligosaccharides, GI mucins, and non-digestible dietary polysaccharides, as well as lactate and formate [91,134]. In the intestines of healthy adults, the ratio of acetate, butyrate, and propionate is maintained around 60:20:20 [143]. However, the level of production of certain SCFAs during the first year of life depends more on the type of diet (breastfeeding or formula feeding), the time of introduction of complementary foods, and the type of bacteria colonizing the infant’s gut [144]. Generally, breastfed infants have been demonstrated to have significantly lower levels of propionate and butyrate before weaning than acetate and higher levels of lactate and formate than after weaning [144]. The high levels of acetate and lactate reflect the dominance of *Bifidobacterium* spp. and *Lactobacillus* spp. [87]. The level of various SCFAs produced by the gut microbiota is of great importance in light of the epigenetic regulation of host genes through modifications to the DNA or histone, and the regulation of non-coding RNAs [145]. Butyrate, propionate, and acetate are not only used locally by GI cells; rather, they have also been found in the peripheral circulation and cerebrospinal fluid, thereby reaching the brain because of their ability to passively or actively traverse the BBB [99,129,143]. SCFAs are absorbed in the brain by glia and, to a lesser extent, by neurons and are considered the main source of their energy, especially at the early stage of brain development [134]. These acids have various effects on the regulation of neurotransmitter levels [99]. Reduced levels of butyrate, propionate, and acetate and impaired dopamine metabolism characterized by decreased levels of HVA have been observed in children with ASD [72].

Propionate, combined with butyrate, can stimulate dopamine and noradrenaline synthesis by enhancing the transcription of the tyrosine hydroxylase gene and promote dopaminergic function by inhibiting the expression of dopamine-β-hydroxylase, which catalyzes the conversion of dopamine to noradrenaline. However, butyrate modulates the expression of genes related to 5-HT production (with a slow onset but long-lasting antidepressant effect), whereas propionate alters the expression of genes related to noradrenaline production (with a fast-acting but short-lived antidepressant effect). Sodium butyrate also facilitated the elimination of hippocampal neuronal abnormalities, increased BDNF gene expression, activated the expression of tight contact protein genes in the BBB, and reduced neuroinflammation by enhancing NMDA receptor activity [99]. SCFAs produced by *Clostridium butyricum* and *B. thetaiotaomicron* have been reported to reduce BBB permeability by increasing the expression of the occlusion-tight contact proteins zonulin and claudin-5 [72].

Another butyrate-mediated effect is an increase in GLP-1 receptor (GLP-1R) expression in the brain and stimulation of GLP-1 production by enteroendocrine cells. The support of appropriate GLP-1 levels by butyrate-producing microbiota in turn contributes to the rise of BDNF in the brain via vagus nerve stimulation [131].

One of the manifestations of ASD is hyperactivity, gait disturbances, and stereotypic movements. The increased incidence of seizure disorders in ASD and other conditions is thought to be related to elevated propionic acid levels. It has been shown that a single injection of propionic acid into rats resulted in ASD-like abnormalities disappearing as it metabolically disintegrated. It has been shown that a single injection of propionic acid into rats resulted in ASD-like abnormalities disappearing as it metabolically disintegrated. Repeated infusions of propionic acid result in neuropathological changes in fatty acid transport in the brain several hours or days after its administration, which occur much later than the transient behavioral effects induced within minutes and lasting about 20–30 min. Feeding formula instead of breastfeeding increased propionic acid levels in the colonic contents of infants. In addition, the use of antibiotics increases the levels of clostridia species that are resistant to them through the spore formation of propionate-producing clostridia [134]. Propionate overproduction by the representatives of the *Bacteroidetes* phylum is also suspected to contribute to the development of AD [146]. Propionate has both anti-inflammatory and pro-inflammatory properties. This compound can influence serotonergic function, which is believed to be involved in anti-inflammatory processes. This is achieved by boosting the activity of tryptophan hydroxylase, the enzyme that transforms tryptophan into 5-HT [99]. According to the report, it has been found that propionate-mediated activation of the free fatty acid receptor 3 (FFAR3) on Schwann cells and dorsal root ganglia and hyperacetylation of histone 3 increase catalase expression, thus promoting resistance to oxidative stress [147]. On the other hand, oral administration of propionate to mice reduced not only 5-HT, but also GABA and dopamine, as well as increased oxidative stress, altered energy metabolism, and increased levels of proinflammatory markers such as IL-6, TNF-α, IFN-γ, heat shock protein 70, and caspase 3 [99]. This compound hindered the action of GABAergic transaminase in neurons, which resulted in an increase in GABA concentration in the brain. Additionally, it also caused a reduction in cerebral glucose metabolism in both the striatum and neocortex of mice [148]. The different outcomes of propionate exposure can be explained by the effects of its different concentration levels. For example, intravenous administration of different propionate doses to rats (2/20/200 mg/kg body weight/day) subjected to chronic, unpredictable, moderate stress alters hippocampal metabolomics. Low doses of propionate effectively rebalance the levels of NTs and NT precursors, norepinephrine, dopamine, GABA, kynurenine, HVA, 3-hydroxyanthranilic acid, 3-hydroxykynurenine, 3,4-dihydroxyphenylacetic acid, and 3-methoxytyramine, and melatonin, that were previously out of balance. However, high doses of the same compound disrupt the metabolism of neurotransmitters and cause changes in the levels of tryptophan, GABA, and 3-hydroxykynurenine, as well as substrates and intermediate products of neurotransmitter metabolism. Oddly, levels of 5-HT did not seem to be affected. The effect of moderate doses was much more complicated, which suggests disturbances in different neurotransmitter-related metabolic pathways [149].

The study of the impact of physiologically relevant doses of SCFAs demonstrated their sex-dependent impact on gene expression in primary cortical astrocytes. Butyrate (0–25 μM) correlates with increased Bdnf and peroxisome proliferator-activated receptor-gamma coactivator (PGC)-1alpha (Pgc1-α) expression in female astrocytes only. Acetate (0–1500 μM) correlates with increased aryl hydrocarbon receptor (Ahr) and glial fibrillary acidic protein (Gfap) expression, and propionate (0–35 μM) correlates with increased Il-22 expression in male astrocytes only [150].

The role of lactate produced by a number of intestinal bacteria in the regulation of neurotransmitter metabolism has not been investigated. Nevertheless, there is evidence that *Bifidobacterium* spp. and *Lactobacillus* spp. dominate in breastfed infants during the first year of life produces L-lactate besides D-lactate [151]. Lactate, along with acetate, is the predominant acid produced during this period until the numbers of *Bacteroidetes* and *Clostridium* representatives increase as the microbiota mature [152]. Research conducted on children from birth until the age of four showed that their urinary excretion of D-lactate was highest during the first year of their lives. However, it decreased by the time they reached the age of 2.5 years and then remained constant [153]. The potential effects of L-lactate produced by bacteria in the first year of life on the brain warrant further investigation. It is reasonable to assume that it functions similarly to how L-lactic acid produced in the muscles during exercise can penetrate the BBB to enhance the synthesis of BDNF [154].

Overall, an imbalance of metabolites produced by the intestinal microbiota following dysbiosis can trigger the development of various psychiatric and neurodegenerative diseases with manifestations of motor dysfunction [128,139].

## 8. Gut Microbiota, Diet and Exercise

As we age, various external factors like diet, stress, medications, and toxins from food cause changes in the microbiota that affect communication between the gut and brain [130]. Therefore, it is essential to pay attention to these factors in addition to age-related changes. By doing so, we can maintain a healthy and well-functioning microbiota, which is crucial for our overall physical and mental health.

The evidence demonstrating the positive therapeutic effects of probiotics in various diseases, including neurodegenerative diseases, has been rising recently. However, their application still depends on careful clinical studies. Significant benefits can be obtained by efficiently improving cases of dopaminergic deficiency in both the CNS and peripheral regions through dietary intervention that regulates dopamine production in the gut microbiota [131]. Studies analyzing the impact of both genetic and dietary factors on the development of metabolic diseases have confirmed that diet plays a primary role in this process. The findings underscore the powerful influence of dietary habits and altered gut microbiota on metabolic health [155]. It has been discovered that the location of a person’s residence plays a crucial role in determining the types of enterotypes that develop in their body. This is due to the traditional dietary patterns of the region [156]. Enterotypes in humans are usually primarily attributed to diet and genetics [157]. However, neonatal bacteria are derived from maternal inheritance [105]. Therefore, when limiting their diet to only breast milk and keeping their gut microbiome simple, we can observe a limited number of enterotypes that are probably determined by inherited core species [158]. And the presence of these enterotypes determining the features of metabolic processes in the gut microbiota probably also contributes to the clinical manifestations of ITW, as observed in ADHD. Subsequently, an enterotype mostly constituted by *Bacteroides* species develops in those who consume a diet high in proteins and animal-based fats, whereas a different enterotype with species from the genus *Prevotella* develops in individuals who follow plant-based diets [156]. Representatives of these two bacterial enterotypes might have different effects on dopaminergic signaling, as discussed above. The timing of complementary feeding and artificial nutrition can have a significant impact on the regulation of NT production by altering the ratio of SCFA to lactate. Therefore, it is crucial to consider the onset of these feeding practices to optimize the production of neurotransmitters in the body. At the same time, diets based on complex sugars are associated with higher levels of *Bifidobacterium* and *Ruminococcus*, and they help regulate dopamine activity. On the other hand, diets that contain a lot of glucose or fructose lead to lower levels of *Bacteroidetes* and higher levels of *Proteobacteria*. The increased production of BDNF in peripheral tissues during a diet high in fiber is most probably associated with the production of butyrate during fermentation by intestinal bacteria, including *Bacteroidetes* [131]. However, recent work has linked increased production of BDNF in adult rats on a pectin-containing diet not to butyrate or propionate but to acetate [159]. Prolonged diets high in sucrose negatively affect dopamine levels in the medial prefrontal cortex because of reduced tyrosine hydroxylase levels [131]. A high-protein diet’s beneficial effects might be attributed to the increased availability of tyrosine, used for dopamine synthesis. In addition, the presence of omega-3 in the diet allows normalization of the *Firmicutes*/*Bacteroidetes* ratio by increasing the level of *Bacteroidetes*, while the level of *Lactobacillus* is also increased [131]. Research on the influence of omega-3 and protein, subject to age restrictions, reveals that their insufficiency during adolescence elevates dopamine release in the dorsal striatum but not in the ventral striatum [160]. This is due to an increased expression of tyrosine hydroxylase. The studies were conducted on humans and male rats of varying ages [131,161]. Although these studies focused on adolescence, there is a high probability that younger children may also have diet-related dopamine neurotransmission abnormalities.

The use of propionate as a food preservative in the production of foods has led to unintended results in our diets. It can be linked to our diet and its impact on our health. This compound, used to prevent mold growth, has become present in the food we consume and can influence our body’s microbiome. Also, widely used in the food industry is propylene glycol (1,2-propanediol), or E1520, which is used as a stabilizer in a number of products. A number of intestinal microorganisms (i.e., *B. thetaiotaomicron*, *Roseburia inulinivorans*, *Escherichia coli*, and *B. breve*) can produce 1,2-propanediol during the fermentation of rhamnose or fucose, whereas *Eubacterium hallii* and *L. reuteri* can metabolize it to propionate [162]. The conversion of 1,2-propanediol to propionate is an important mechanism for regulating the maintenance of the ratio between SCFAs. Consuming propionate- or 1,2-propanediol-containing products can potentially worsen SCFA’s balance in children’s guts. It may lead to an increase in propionate levels and slight disturbances in neurotransmitter production within the brain. In addition, industrial food production leads to food also containing advanced glycation products (AGEs), which have toxic effects on the intestinal microbiota and disrupt immune homeostasis [163]. The existence of AGEs in food is considered to change the biomechanical plasticity of tendons, in particular the Achilles tendon [164,165]. A hypothesis was also proposed that shortening of the Achilles tendon in the case of diabetes mellitus occurs because of collagen fiber destruction after AGE accumulation [166]. The development of tendon connective tissue stiffness also happens because of gut dysbiosis, which initiates inflammatory arthropathies by triggering immunological processes [167]. The disappearance of several *Bacteroides* species, particularly *B. uniformis* and *Bacteroides plebeius*, in the first stage of rheumatoid arthritis impairs glycosaminoglycan metabolism and triggers articular cartilage damage that continues in the later stages of the disease [168]. Furthermore, glucosamine can inhibit glucose uptake in differentiated NSC-34 motor neuron cells in vitro, consequently reducing the level of intracellular ATP. GLP-1’s restorative effect on glucose uptake [169] has been shown to protect motor neurons, indicating that disrupted bacterial glucosamine metabolism may have varying consequences for the host organism. The microbiota has been found to produce positive changes in the immune profile, which leads to an improved healing process for the Achilles tendon in rats. This further demonstrates the interconnectedness between the microbiome and the body’s immune response. Studies show that the effect is significant, indicating the potential of the microbiota to treat similar injuries in humans [170,171]. Thus, additional research is also required on the involvement of the intestinal microbiota of children with IWT in the development of Achilles tendon stiffness. The development of Achilles tendon stiffness and shortening in several children with ITW, together with habitual use of the gastrocnemius muscle and changes in the acting load, can lead to remodeling of the muscle-tendon architecture and the formation of equinus contracture [172,173,174].

Regular physical exercise is not only essential for maintaining good health, but it can also help restore various bodily functions that have been impaired by chronic medical conditions. Furthermore, shoe orthoses represent one of the more conservative approaches to treating ITW [11,14]. Although a small study showed no significant reduction in physical activity among children with ITW, engaging in physical activity can significantly enhance their quality of life [175] and reduce the duration of ITW [176]. Therefore, physical activity should be encouraged and not limited by the condition. Studies have shown that shoe orthoses can help manage ITW, but it is essential to encourage physical activity too, as it can promote better overall health and wellbeing. Regular exercise, whether aerobic or anaerobic, can lower the chances of oxidative brain damage. However, it is vital to ensure that the exercise routine is not too taxing on the body. Keeping physical activities consistent without pushing oneself to the limit is essential for health benefits [132]. Physical exercise also increases the diversity of the gut microbiota, and its effect depends on the type and intensity of exercise [177,178]. Most of the studies on the effects of exercise on the microbiota have been conducted on athletes, so we do not discuss the changes observed. However, interestingly, a study of the effect of moderate exercise in rats revealed an increase in the number of species of *Lactobacillus*, *Bifidobacterium*, and *Blautia coccoides*-*Eubacterium rectale* group, with a simultaneous decrease in the number of genera *Clostridium* and *Enterococcus* compared to sedentary male rats [179]. Recent research has revealed that the existence of *Lachnospiraceae* and *Lactobacillaceae* families in microbiota is negatively linked with endurance performance, whereas the *Prevotellaceae* family, *Prevotella* genus, and *Akkermansia muciniphila* exhibit a positive correlation with resistance performance [178]. Moreover, the effect of exercise on changes in gut microbial communities is more stable compared to dietary effects [179]. Exercise maintains the structural integrity and function of the brain, enhances hippocampal neurogenesis, and increases memory capacity [56]. Various studies conducted both in preclinical models and with the participation of normal individuals and AD and PD patients revealed that aerobic exercise increases mRNA and BDNF protein levels in the hippocampus, which might be associated with an increase in lactate production upon physical exercise [132,154]. Physical exercise also stimulates angiogenesis in the hippocampus by promoting increased production of vascular endothelial growth factor (VEGF) in myocytes, followed by their diffusion into peripheral circuits and overcoming the BBB. Of note, neurogenesis is enhanced under the action of VEGF [132], as well as due to adrenaline [56]. Besides, they facilitate a decrease in the expression of proinflammatory cytokines and an increase in the expression of anti-inflammatory cytokines [56].

## 9. Conclusions

The exact cause of ITW is not yet fully comprehended; however, it typically appears when a child reaches a crucial phase of their growth and development. By this time, their natural systems, including the immune, digestive, nervous, and musculoskeletal systems, have undergone significant maturation. Notably, changes in the composition and metabolism of the gut microbiota are related to this transition because its metabolism is closely related to that of the host.

Our review suggests a new idea based on Bauer et al.’s ITW classification (Figure 1). We propose that the dysbiosis of the intestinal microbiota that disturbs neurotransmitter production may underlie the ITW phenotype in the “Developmental” and “Persistent” ITW groups in children aged 2–3 years. In the case of “Persistent ITW”, both genetic factors and metabolic disorders can influence the development of ITW. The stiffness of the connective tissue and the shortening of the Achilles tendon in this case can also develop because of disturbances in the exchange of connective tissue provoked by dysbiosis, contributing to the development of contracture. Moreover, if abnormal brain circuitry formation persists and is not addressed through gait training, it can deteriorate further due to the prolonged metabolic imbalances resulting from gut microbiota dysbiosis. This can potentially lead to the development of “Persistent Habitual ITW”.

We pose that the presence of dysbiosis, a contributing factor in TW, might also exist in the “TW + ASD” group, which may have different sensory sensitivity [180,181]. This is plausible considering the reported occurrence of various types of intestinal dysbiosis in children with ASD [182]. However, it could potentially have a greater impact, extending beyond sensory processing to affect communication and social interaction. In children who do not have a genetic predisposition, the development of ITW can be avoided by achieving a healthier lifestyle. This includes eating a nutritious diet and engaging in regular physical activity. Moreover, early prophylaxis with probiotics could significantly speed up treatment and help avoid negative outcomes in children with ITW who had risk factors leading to dysbiosis, such as premature birth. However, conducting proper research will be necessary.

## Figures and Tables

**Figure 1 ijms-24-13204-f001:**
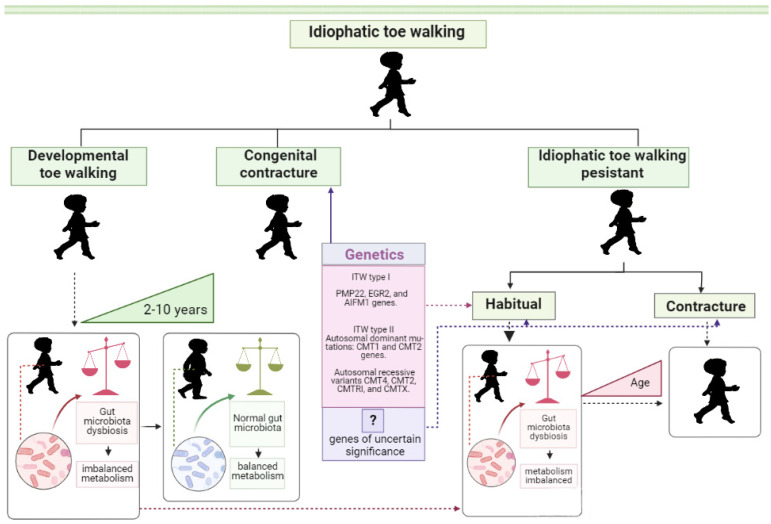
The modifying nomenclature of idiopathic toe walking by Bauer et al. (2022) [15] in accordance with the influence of genetic factors and metabolic disturbances of the intestinal microbiota that modify sensorimotor processing. “Congenital contracture” is determined by unreported genetic mutations. “Developmental ITW” is a transient condition due to the resolution of the metabolic imbalance induced by gut microbiota dysbiosis. Restoration of normal microbiota through diet and exercise leads to the disappearance of ITW. In cases of “Persistent ITW”, as genetic factors as metabolic disturbances can affect the development of ITW. Contracture in this case can also develop as a result of impaired connective tissue metabolism provoked by dysbiosis and as an adaptation to the toe walking pattern. This figure has been designed using an image from BioRender (https://www.biorender.com).

**Figure 2 ijms-24-13204-f002:**
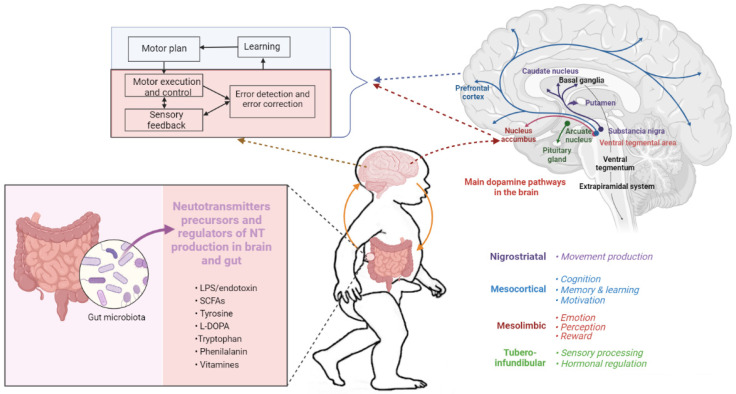
The motor learning of walking in toddlers. Learning to walk is a significant developmental milestone for toddlers. It allows them to explore and interact with their environment in new ways. The process of motor learning involves the complex coordination of different body parts, muscles, and nervous system functions. As toddlers begin to develop the ability to walk, they must learn how to maintain balance and coordinate movement. This process involves trial and error, as well as learning from feedback and experiences. With practice, toddlers begin to develop muscle memory, making walking a more natural and automatic process. It is an exciting and crucial stage in a child’s development, demonstrating their growing independence and ability to navigate the world around them. The brain generates a motor plan for movement. The brain is responsible for controlling movements and utilizing a forward model to anticipate the outcomes of sensory motor commands and make adjustments as needed for precision. The dopamine-producing neurons of the mesocorticolimbic pathway are responsible for movement planning and correction. The gut may serve as a source of NT precursors for the production of dopamine, glutamate, GABA, and 5-HT. These neurotransmitters play a crucial role in the neuroregulation of motility-related processes and can be produced with the help of NT precursors found in the gut. SCFAs and vitamins can modulate NT production; LPS can affect NT production via neuroinflammation. Abbreviations: NT—neurotransmitters; GABA—γ-aminobutyric acid; 5-HT—serotonin; SCFAs—short-chain fatty acids; LPS—lipopolysaccharide. This figure has been designed using images from BioRender (https://www.biorender.com).

## Data Availability

Data sharing not applicable.

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
