# Peer review of "Gut Microbiome Dysbiosis as a Potential Risk Factor for Idiopathic Toe-Walking in Children: A Review"

_ijms, 2023, doi:10.3390/ijms241713204_

Round 1
Reviewer 1 Report (New Reviewer)
Dear Editors and authors ,
1-The title of the manuscript is very pretentious since the text does not show novel aspects and only describes general information on the subject.
2-The summary is very general and does not show information that refers to the importance of the performed review.
3-Regarding the different analyzed sections, a lot of information is included, however, much of it not relevant to the topic. It is recommended that the authors of the manuscript, evaluate writing it again, it is important that they be careful regarding the objective of the study and what it contributes to new knowledge.
4- The title of figure 2 is not correct.
5- Add required bibliography.
6-The conclusions are not good and contain inaccurate and focused information, as well as the presence of an important figure within the conclusions, and this chapter of the manuscript should not contain figures or tables.
The quality of English is good.
Author Response
Please see the attachment

Reviewer 2 Report (New Reviewer)
This is an interesting review discussing the relationship between gut microbiome dysbiosis and idiopathic to-walking in children. I have several comments to improve the manuscript further:
1. The introduction tends to jump between topics related to ITW. For a more coherent flow, start from the general aspects of psychomotor development, then delve into toe-walking, and finally into the main focus of the paper - potential etiologies and the proposed hypothesis.
2. The phrase "Our expectation is" seems unnecessary. Simply state: "This review explores the potential link between microbiome variations and ITW."
3. The authors should consider tone down the title as the current review is just an exploration.
4. While the sections on walking mechanics and brain development are both detailed, there's an opportunity to draw more direct connections between the two. For instance, how do neural networks or synapses specifically support the development of walking? Are there brain areas more activated during the learning phases of walking?
5. It might be beneficial to include typical developmental milestones related to both brain development and walking. This would offer readers a practical framework or timeline.
6. The text states, "Once a child rich the age of 7...". This appears to be a typographical error. "Rich" should probably be "reaches".
7. The discussion about gut microbiota and its influence on the dopaminergic pathway seems suddenly introduced. Some introductory or transitional sentences leading into this topic might be helpful.
8. The section discussing glyphosate, while interesting, might seem tangential without appropriate context. The connection between glyphosate, the gut microbiota, and dopamine might be strengthened with a clearer explanation.
9. The mention of "LPS-induced inflammation" is sudden. Explaining what LPS is (Lipopolysaccharide) and its relevance to the broader discussion might help the reader understand its significance.
10. The connection between different bacteria and their impact on dopamine and related pathways is presented as a series of facts. However, without more context on the studies or potential confounding factors, it's challenging to assess the robustness of these claims.
11. There's a statement about the "development of ASD-like conditions in children due to the chronic presence of glyphosate in the diet." This is a significant claim and needs a strong evidentiary base. The potential relationship between glyphosate and neurodevelopmental disorders is a contentious topic, so it's crucial to ensure that such claims are supported by robust research.
12. The text states that "Despite levels of glyphosate being widely accepted in agriculture and considered safe for health [79], there is evidence of the development of ASD-like conditions in children due to the chronic presence of glyphosate in the diet [78, 80]." These statements appear contradictory. If glyphosate is considered safe for health, why is there evidence suggesting it can lead to ASD-like conditions? This needs clarification.
13. It's important to maintain consistency. For instance, "Bifidobacterium spp." is mentioned, but "Bacteroidetes" is used instead of the specific "Bacteroides spp.".
14. There are claims like "Over the past decades, infant breastfeeding and its duration, especially after cesarean section, have declined dramatically worldwide." It would be important to specify which parts of the world this refers to, as trends can differ significantly between countries and regions.
15. At one point, there is mention of pesticides and herbicides (specifically glyphosate) as causes of intestinal microbiota dysbiosis. While there is research on this topic, drawing a direct line between these chemicals and human gut dysbiosis requires more nuance. Environmental, dietary, and genetic factors can play roles.
16. Some links between the gut microbiota and neurological development seem to be correlations, not necessarily causations. It's vital to clarify this to avoid misleading interpretations.
17. Please ensure consistent use of terminology. For example, both 5-HT and serotonin are used. While many in the field will recognize these as the same, it can be confusing for some readers.
18. When discussing how the gut neurotransmitters influence the brain, more clarity on the mechanisms might be helpful. For instance, how do they modulate the immune system, and how does that, in turn, affect the brain?
19. The section about propionate's dual roles – its anti-inflammatory and pro-inflammatory properties – may need to be elaborated on more. Why does propionate have these seemingly contradictory roles? Under what conditions does each take precedence?
20. Phrasing such as "The idea of the effect of L-lactate produced by bacteria in the first year of life on the brain sounds very attractive" sounds a bit informal for a scientific review. It may be more appropriate to phrase it objectively, e.g., "The potential effects of L-lactate produced by bacteria in the first year of life on the brain warrant further investigation."
21. The connection between gut microbiota and ITW is presented as a novel idea. However, it's essential to make clear that this is a hypothesis or emerging area of research. The direct causative link between gut microbiota dysbiosis and ITW is still not well-established.
22. The text indicates that propionate has unintended consequences in our diets and impacts our gut microbiome. While this may be true, the exact ramifications of consuming propionate-containing products could be elaborated upon further.
23. Lastly, given that this is a review, it's essential to present a balanced view, including studies or opinions that might not align with the proposed hypothesis. This balance ensures that the review is comprehensive and fair.
Round 2
Reviewer 1 Report (New Reviewer)
Dear Editors,
The authors have made all required modifications. The manuscript is now ready for publication.
Author Response
Many thanks
Reviewer 2 Report (New Reviewer)
The authors have addressed my comments well. I have only one further comment which is related to the title. I feel that the authors should clearly indicate that it is a review or an exploratory review in the title.
Author Response
We appreciate the Reviewer's proposal, the title of the article has been changed.
This manuscript is a resubmission of an earlier submission. The following is a list of the peer review reports and author responses from that submission.
Round 1
Reviewer 1 Report
The authors outline what is known about idiopathic toe walking. Then outline what is known about gut health.
Without evidence, what is then attempted is linking the two in a theoretical way. This condition (ITW) is benign in 80% of the cases. The authors have not provided any evidence to support this theory. In the absence of this, the danger this article could do is further escalate the concerns of parents and health seeking behaviours.
If the researchers are committed to this linkage, rather than theorising, potentially contributing to health wastage, I would recommend a case series or trial to better understand if there is a link in some children. In its present form, it does not contribute to what is known about idiopathic toe walking.
I do not recommend this be published based on the limited evidence and weak (or no) links the authors have outlined.
Level of English is acceptable.
Author Response
Response to Reviewer 1 Comments
Point 1: This condition (ITW) is benign in 80% of the cases.
Response 1: We agree that in the orthopedic literature, ITW has been described as an isolated benign sign. However, a number of references do report a few children with TW and learning problems. Moreover, given the large number of children with ITWp presenting for evaluation in orthopaedic practice, it is evident that not all children resolve spontaneously. Little is written on the longterm sequela of untreated TW; however, abnormal talar morphology, including the lack of a talar recess, has been documented in 66% of children with ITW and loss of talar roundness in 30% of children with ITW. Persistent TW in children older than 3 years often results in parental concern, provoking multiple medical visits, and a range of interventions including physical therapy (PT), stretching (SC), botulinum toxin A (BTX-A) injections, and surgery. Nearly 59% and 80% of children who toe walk as a phase of normal gait development during early development [6] recover normal walking (a heel‐toe gait) by the age of 5,5 [7] and 10 [8]years, respectively. However, there are cases in which TW persists in children, which might be unilateral or even developed after a period of heel walking [9].
Point 2: The authors outline what is known about idiopathic toe walking. Then outline what is known about gut health. Without evidence, what is then attempted is linking the two in a theoretical way. The authors have not provided any evidence to support this theory.
Response 2: Respecting the Reviwer’s point, we assumed that our review does not completely reach it’s goal , i.e., to establish a strong theoretical foundation and highlight the benefits of further exploring the possible connection between alterations in the microbiome and ITW. Hence, we rewrite (changes marked yellow in the revised version of our manuscript) the introduction and several other sections to substantiate this hypothesis using the data obtained in the studies of several pathologies manifested by association of sensory-motor dysfunction and intestinal dysbiosis.
On this matter, we would mention also that a number of studies have also indicated a significant link between ITW in children, their family's history, and sensory processing difficulties along with autism/autism spectrum disorders (ASD) and speech development disorders [10; Haynes KB, Wimberly RL, VanPelt JM, Jo CH, Riccio AI, Delgado MR. Toe Walking: A Neurological Perspective After Referral From Pediatric Orthopaedic Surgeons. J Pediatr Orthop. 2018 Mar;38(3):152-156. doi: 10.1097/BPO.0000000000001115. PMID: 29309384; Valagussa G, Purpura G, Nale A, Pi-rovano R, Mazzucchelli M, Grossi E, Perin C. Sensory Profile of Children and Adolescents with Autism Spectrum Disorder and Tip-Toe Behavior: Results of an Observational Pilot Study. Children (Basel). 2022 Sep 1;9(9):1336. doi: 10.3390/children9091336. PMID: 36138645; 20; 26]. Moreover, numerous research studies have highlighted the presence of altered sensory processing, encompassing tactile, proprioceptive, vestibular, and visual processing, in several children affected by ITW. Furthermore, multiple initiatives were implemented to confirm the disruption of sensorimotor regulation. [21, 26, 27, 28, 29, 30, 31].
Although ITW has been known for more than half a century, we still don't know much about its etiology. Currently, there is a growing interest in the diagnosis, treatment, classification, and causes of ITW [DOI: 10.1055/s-0042-1742583, 22]. This is due in part to the fact that conservative treatment of persistent ITW often yields unsustainable results.
From the nomenclature proposed by Bauer et al., it follows that sensorimotor disor-ders are associated with ASD [22]. Sensorimotor impairment has been reported in ITW in a number of children. ASD belongs to a group of neurodevelopmental disorders [35447717] that can manifest with sensorimotor impairments. However, TW is one of the various symptoms that can be encountered in other types of disorders as well. Although it may not seem immediately relevant, the categorization of "ITW + ASD" is actually very significant because it highlights the well-known connection between ASD and an imbal-ance in the neurotransmitter system that is caused by dysbiosis of the gut microbiome [Golubeva AV, Joyce SA, Moloney G, Burokas A, Sherwin E, Arboleya S, et al. Microbio-ta-related changes in bile acid & tryptophan metabolism are associated with gastrointes-tinal dysfunction in a mouse model of autism. EBioMedicine. (2017) 24:166–78; Agus A, Planchais J, Sokol H. Gut microbiota regulation of tryptophan metabolism in health and disease. Cell Host Microbe. (2018) 23:716–24. 10.1016/j.chom.2018.05.003; Gheorghe CE, Martin JA, Manriquez FV, Dinan TG, Cryan JF, Clarke G. Focus on the essentials: trypto-phan metabolism and the microbiome-gut-brain axis. Curr Opin Pharmacol. (2019) 48:137–45. 10.1016/j.coph.2019.08.004; Xu M, Xu X, Li J, Li F. Association between gut mi-crobiota and autism spectrum disorder: a systematic review and meta-analysis. Front Psychiatry. (2019) 10:473. 10.3389/fpsyt.2019.00473; Fattorusso A, Di Genova L, Dell’Isola GB, Mencaroni E, Esposito S. Autism spectrum disorders and the gut microbiota. Nutri-ents. (2019) 11:521. 10.3390/nu11030521; Sharon G, Cruz NJ, Kang D-W, Gandal MJ, Wang B, Kim Y-M, et al. Human gut microbiota from autism spectrum disorder promote behavioral symptoms in mice. Cell. (2019) 177:1600–18.e17. 10.1016/j.cell.2019.05.004]. The imbalance occurring in gut microbial communities and their metabolites has a sig-nificant effect on the emergence of ASD in children and animal models [Tabouy L, Get-selter D, Ziv O, Karpuj M, Tabouy T, Lukic I, et al. Dysbiosis of microbiome and probiotic treatment in a genetic model of autism spectrum disorders. Brain Behav Immun. (2018) 73:310–9. 10.1016/j.bbi.2018.05.015 ]. Research has demonstrated that the microbial com-position present in mothers can significantly influence the neurological development of their offspring in the future [Azad MB, Konya T, Maughan H, Guttman DS, Field CJ, Chari RS, et al. Gut microbiota of healthy Canadian infants: profiles by mode of delivery and in-fant diet at 4 months. Can Med Assoc J. (2013) 185:385. 10.1503/cmaj.121189; Shafai T, Mustafa M, Mulari J, Curtis A. Impact of infant feeding methods on the development of autism spectrum disorder. In: Fitzgerald M, Yip J editors. Autism – Paradigms, Recent Research and Clinical Applications. London: IntechOpen; (2017)]. Revisiting Bauer et al.'s proposed ITW classification [22] through the lens of genetic disorders and subclinical conditions in sensorimotor control suggest that an imbalance in metabolic regulation of neurotransmitter synthesis due to dysbiosis of the gut microbiota may account for "De-velopmental ITW" and partially for "Persistent ITW". There is overwhelming evidence to support the similar mechanism of the significant impact of gut microbiota in the devel-opment of various neurodevelopmental and neurodegenerative conditions [reviewed in Mitrea L, Nemeş SA, Szabo K, Teleky BE, Vodnar DC. Guts Imbalance Imbalances the Brain: A Review of Gut Microbiota Association With Neurological and Psychiatric Disor-ders. Front Med (Lausanne). 2022 Mar 31;9:813204. doi: 10.3389/fmed.2022.813204; Wang X, Ma R, Liu X, Zhang Y. Effects of long-term supplementation of probiotics on cog-nitive function and emotion in temporal lobe epilepsy. Front Neurol. 2022 Jul 19;13:948599. doi: 10.3389/fneur.2022.948599; Dickerson F, Dilmore AH, Godoy-Vitorino F, Nguyen TT, Paulus M, Pinto-Tomas AA, Moya-Roman C, Zuniga-Chaves I, Severance EG, Jeste DV. The Microbiome and Mental Health Across the Lifespan. Curr Top Behav Neurosci. 2023;61:119-140. doi: 10.1007/7854_2022_384; Góralczyk-Bińkowska A, Szmajda-Krygier D, Kozłowska E. The Microbiota-Gut-Brain Axis in Psychiatric Disor-ders. Int J Mol Sci. 2022 Sep 24;23(19):11245. doi: 10.3390/ijms231911245; Qin D, Ma Y, Wang Y, Hou X, Yu L. Contribution of Lactobacilli on Intestinal Mucosal Barrier and Dis-eases: Perspectives and Challenges of Lactobacillus casei. Life (Basel). 2022 Nov 16;12(11):1910. doi: 10.3390/life12111910; Hong D, Zhang C, Wu W, Lu X, Zhang L. Mod-ulation of the gut-brain axis via the gut microbiota: a new era in treatment of amyotrophic lateral sclerosis. Front Neurol. 2023 Apr 20;14:1133546. doi: 10.3389/fneur.2023.1133546; Turroni S, Provensi G. Editorial: Gut biodiversity and its influence in brain health. Front Neurosci. 2023 Jun 1;17:1221543. doi: 10.3389/fnins.2023.1221543]. That may serve as in-direct confirmation of the relevance of such an assumption. Of note, at present, there are no definitive and effective therapies for the treatment of ASD, nor patients of "ITW + ASD" category. Households can face this situation by implementing various strategies such as adding supplements like vitamins, pre- and pro-biotic formulations, and multi-minerals. They can also undergo specifically tailored dietary, psycho-pharmacological, and educational treatments.
Indeed, children with ADHD or ASD has frequently gut microbiota dysbiosis [38-40], which may be involved in the pathogenesis of the disease via immune and inflammatory mechanisms [41-43]. Pre-cognitive impairment motor dysfunction in neurodegenerative diseases, for example, Alzheimer's disease (AD) and Parkinson's disease (PD), has also been associated with dysbiosis of the gut microbiota [44-46]. The initiation of neurodegen-erative disorder in PD is assumed to begin with α-synuclein aggregation in the enteric nervous system as early as the asymptomatic premotor stage of the disease, followed by the spread of gut-related inflammation to the CNS. Development of degeneration of do-paminergic systems in the compact part of the substantia nigra and abnormal aggregation of α-synuclein in the remaining dopaminergic neurons leads to a deficit of executive func-tion (information processing speed and working memory), causing postural instabil-ity/gait disturbances [47, 48]. Degeneration within the cholinergic systems seems to favor axial motor symptoms, particularly, slow walking speed and loss of balance in individu-als with PD [47, 48]. Both patients with PD and children with ASD have reduced levels of Prevotella copri and increased levels of family Enterobacteriaceae, of which the latter is asso-ciated with gait disturbances and postural instability in PD [49].
Based on the analysis of literature data, in this review we pose and substantiate the novel hypothesis that the disturbances of neurotransmitter production as a result of dysbiosis of the intestinal microbiota at the age of 2-3 years may underlay the expression of the ITW phenotype in the “Developmental” and “Persistent” ITW groups. Our hypothesis suggests that dysbiosis may also be observed in the "ITW + ASD" group, however, the impact might be much more extensive, affecting not only sensory processing but also communication and social interaction.
Point 3: In the absence of this, the danger this article could do is further escalate the concerns of parents and health seeking behaviours.
Response 3: From the medical ethics point of view and being doctors, most of us would disagree that the escalation of parent's concerns and healh seeking behaviours can be considered as any danger for true patient-centered clinicians.
Once again, we would emphasize that this review is addressed to scientific community and aimed to establish a strong theoretical foundation and highlight the benefits of further exploring the possible connection between alterations in the microbiome and ITW.
Point 4: If the researchers are committed to this linkage, rather than theorising, potentially contributing to health wastage, I would recommend a case series or trial to better understand if there is a link in some children.
Response 4: Appreciating an excellent Reviewer’s suggestion, we would point out that we are fully aware of the necessity of validation of a definitive association between gut microbiota dysbiosis and sensorimotor impairment in infants with ITW. We understand that it cannot be established without longitudinal multi-center studies involving a significant number of children. Therefore, our review boldly advocates for continued research into the potential link between microbiome variations and ITW. We believe that validated assumptions offer hope for the ITW symptoms to be cured in the initial stage, before irreversible morphological lesions occur.
Point 5: In its present form, it does not contribute to what is known about idiopathic toe walking. I do not recommend this be published based on the limited evidence and weak (or no) links the authors have outlined
Response 5: We hope that we addressed this issue in our Response 2 and within the context of our new revised version of the manuscript. We sincerely hope that our revised version meets the high standards of The International Journal of Molecular Sciences and that it is considered worthy of publication in such a prestigious and influential journal.
Please see the attachment, our arises are highlighted in yellow

Reviewer 2 Report
Some comments given to the auhtros.
1. As the last note in your abstract, please provide a "take-home" message.
2. Keywords should be reordered based on alphabetical order.
3. What is the novel bought by the authors in the current submission? Its review works have been widely discussed in the past. Nothing something really new in the present form. The lack of a novel seems to make the present submission like to replication/modified review. The authors need to detail their novelty in the introduction section. It is a major concern for rejecting this paper.
4. In order to highlight the gaps in the literature that the most recent literature aims to fill, it is crucial to review the benefits, novelty, and limitations of earlier reviewin the introduction.
5. Explain specifically the objective of the present review in the last paragraph of the introduction section.
6. Encouraging to the authors to provide an additional figure in the introduction section for increasing the quality of the present submission.
7. More explanation of ITW still needed, the present form was unclear.
8. Autism Spectrum Disorder (ASD) refers to a group of complex neurodevelopmental disorders that begin in the developmental period and is indicated by deficits in communication and social interaction as well as restricted and repetitive interest behaviors that are usually recognized in early childhood with the presence of two major symptoms: social– communication deficits and restricted and repetitive interests/behaviors. Please provide general explanation of ASD in the manuscript and refer the relevant study as follows: https://doi.org/10.3390/bioengineering9040157
-
Author Response
Point 1: As the last note in your abstract, please provide a "take-home" message.
We modify our abstract to provide a "take-home" message
Point 2: Keywords should be reordered based on alphabetical order.
Keywords were reordered
Point 3: What is the novel bought by the authors in the current submission? Its review works have been widely discussed in the past. Nothing something really new in the present form. The lack of a novel seems to make the present submission like to replication/modified review. The authors need to detail their novelty in the introduction section. It is a major concern for rejecting this paper.
and
Point 4: In order to highlight the gaps in the literature that the most recent literature aims to fill, it is crucial to review the benefits, novelty, and limitations of earlier reviewin the introduction.
and
Point 5: Explain specifically the objective of the present review in the last paragraph of the introduction section.
We have modified the introduction to the manuscript to focus more on the problem of the etiology of ITW, which is currently understudied. Recent reviews discuss only a few mutations that explain ITW. Sensorimotor processing impairments and a high incidence of children with ITW have also been reported, in whom the diagnosis was most often refined over time as ASD. Based on these data, we reviewed the new nomenclature of the disease proposed in the review by Bauer et al [15] and hypothesized that in the “Developmental ITW” and “Persistent ITW” groups proposed by him, the cause of ITW development is intestinal microbiota dysbiosis. Previously, the role of intestinal microbiota dysbiosis for these groups of children was not discussed by anyone. The metabolic imbalance caused by dysbiosis is lower than in the case of the "ITW + ASD" group, so children can show typical development.
P.s. Many thanks for your point 4, it helped improve the introduction
To clarify the role of individual members of the microbiota in children in the development of motor dysfunction, we have added a chapter on gut colonization.
We have also added several new references further supporting the effect of microbiota members and their metabolites on sensorimotor function in children.
We also reworked the conclusion in the context of the hypothesis under discussion and added a figure.
All edits are highlighted in yellow. The numbering of links has also been corrected, taking into account the references made.
Point 6: Encouraging to the authors to provide an additional figure in the introduction section for increasing the quality of the present submission.
We added the figure in conclution
Point 7: More explanation of ITW still needed, the present form was unclear.
We corrected the term of ITW in the introduction
Point 8: Autism Spectrum Disorder (ASD) refers to a group of complex neurodevelopmental disorders that begin in the developmental period and is indicated by deficits in communication and social interaction as well as restricted and repetitive interest behaviors that are usually recognized in early childhood with the presence of two major symptoms: social– communication deficits and restricted and repetitive interests/behaviors. Please provide general explanation of ASD in the manuscript and refer the relevant study as follows: https://doi.org/10.3390/bioengineering9040157
We provided general explanation of ASD in context of new version of the introduction
Please see the attachment

Round 2
Reviewer 1 Report
While I appreciate that the authors have spent time attempting to strengthen the links, I am still challenged by this paper.
There are fundamental flaws that I don’t believe are addressed during review , in fact in fact have weakened the link, exposing greater flaws in the argument.
key issues:
1. ITW is not linked with autism. iTW by nature is lack of a condition know to cause or be associated with toe walking. This means toe walking and autism is not itw. Same was toe walking and DCD, global developmental delay etc, not itw, but toe walking associated with neurogenic or developmental conditions.
2. Not all children who have autism, toe walk, approx 4 in 7 do.
3. This means if there is a gut biome linked with autism, it is unknown if linked with toe walking as not all children with autism who toe walk. If your argument (and proof of concept) was that all kids who have autism and gut issues toe walk, I’d be willing to accept this hypothesis to other causes of toe walking, but to then stretch to ITW is too far.
4. Other studies in ITW have linked changes in tone or even muscle fibre with ITW, pointing to an underlying subtle neuro cause or genetic causes given it commonly runs in families, but no one is willing yet to push that, because ITW is too variable.
5. I’d urge is authors are committed to this theory, work on linking toe walking and autism and gut health prior to then stepping out to ITW, children with itw are too diverse a population and hypothesis too far from anything right now to be believed without creative more stress on parents.
Nil
Author Response
Point 1: ITW is not linked with autism. iTW by nature is lack of a condition know to cause or be associated with toe walking. This means toe walking and autism is not itw. Same was toe walking and DCD, global developmental delay etc, not itw, but toe walking associated with neurogenic or developmental conditions.
Response 1: We fully agree with you and wrote in the article that ITW is not linked with autism, albeit Bauer et al. included a separate group of "autism+ ITW" in the ITW classification. Moreover, we emphasized that this contradicts the very definition of ITW, and we do not even mention this group on Fig. 3, where we tried to give a graphical representation of our hypothesis. Indeed, the nature of ITW is not clear, nor are the conditions causing or associated with toe walking in autism or DCD. We do not pretend anything contrary to what you write, and in order to avoid confusion, we clarify this part of the introduction by specifying this group as "autism + TW".
Bauer et al. Idiopathic Toe Walking: An Update on Natural History, Diagnosis, and Treatment. Am. Acad. Orthop. Surg. 2022, 30, e1419-e1430.
Our correction: line 109-110, "TW + ASD" in line 111 and 122
Point 2: Not all children who have autism, approx 4 in 7 do.
Response 2: Indeed, the autism group is heterogeneous, and only a fraction of patients have a toe walk. This issue is actively discussed in the literature. This is not to say that all researchers are unanimous in the causes of this phenomenon. Most, including us, are of the view that with autism there is an alteration of sensory perception.
More recently, Valagusa et al. have analysed different modes of sensory sensitivity in children with ASD and the presence or absence of TW. Based on the experimental data obtained, they showed that the “autism + TW” group compared to the autistic group without TW was characterized by a more pronounced decrease in the perception of sensory stimuli.
Valagussa G, Purpura G, Nale A, Pirovano R, Mazzucchelli M, Grossi E, Perin C. Sensory Profile of Children and Adolescents with Autism Spectrum Disorder and Tip-Toe Behavior: Results of an Observational Pilot Study. Children (Basel). 2022 Sep 1;9(9):1336. doi: 10.3390/children9091336.
Point 3: This means if there is a gut biome linked with autism, it is unknown if linked with toe walking as not all children with autism who toe walk. If your argument (and proof of concept) was that all kids who have autism and gut issues toe walk, I’d be willing to accept this hypothesis to other causes of toe walking, but to then stretch to ITW is too far.
Response 3: If "autism + TW" is significantly different from the "autism without TW" group, it is reasonable to assume that they differ in the balance of neurotransmitters that provide sensory perception. At present, there is no doubt about the effect of the intestinal microbiome on the status of neurotransmitter systems. It is likely that the microbiome is also different in these two groups of autistic patients. Furthermore, there is a correlation between the degree of change in sensory perception and the degree of expression of TW, and thus it may correlate with changes in the microbiome. Hence, we hypothesize that the presence of two groups of autistic patients may be fully explained by changes in the microbiome. The question of altering the balance of neurotransmitters and dysbios in children with ITW has not yet been raised.
Differences in the microbiomes of ASD patients are summarized in a review by Plaza-Diaz et al.. As can be seen from the review, there are different types of intestinal dysbiosis in ASD, both with varying degrees of abundance of bifidobacteria and lactobacilli, and with different ratios of Firmicutes/Bacteroidetes and Bacteroidetes/Firmicutes. The review also notes that several studies have shown that:
- some neurotransmitters involved in ASD are regulated by the microbiome, including glutamate, serotonin, and dopamine;
- children with ASD had significantly lower levels of SCFAs, propionic, acetic and butyric acids, as well as a hyperserotonergic state (increased serotonin levels) and impaired dopamine metabolism (decreased homovanillic acid levels). Administration of probiotics in combination with fructooligosaccharides resulted in SCFA levels close to those found in the control group, along with a concomitant decrease in serotonin levels and an increase in homovanillic acid levels;
- in treatment with prebiotics and synbiotics, changes in microbial metabolism have been associated with an increase in the concentration of SCFAs and a decrease in ammonium levels;
- A study of metabolites in urine of ASD patients demonstrate dysregulation of the purine, tyrosine, phenylalanine, and tryptophan pathways, characterized by an increase in phenylalanine levels and a decrease in tyrosine levels. In addition, there was a significant increase in the concentration of bacterial metabolites: phenylacetic acid, phenylpyruvic acid and 4-ethylphenyl sulfate.
We've also added to Conclusions this clarification Line 1013-1016.
Besides we've also added to Part 6.3. clarification about vaginal dysbiosis in mothers of preterm babies, which is believed to provoke premature birth. Alterations in the composition of Lactobacilli transmitted to the baby at birth may also cause dysbiosis in children with ITW. Line 545-551.
We once again thank Reviewer for the very important questions raised, which, in our opinion, once again emphasize the need to study the alteration of bacterial metabolism in both children with ITW and children of the "autism + TW" group.
Plaza-Diaz, J., Radar, A. M., Baig, A. T., Leyba, M. F., Costabel, M. M., Zavala-Crichton, J. P., ... & Solis-Urra, P. (2022). Physical Activity, Gut Microbiota, and Genetic Background for Children and Adolescents with Autism Spectrum Disorder. Children, 9(12), 1834
Point 4: Other studies in ITW have linked changes in tone or even muscle fibre with ITW, pointing to an underlying subtle neuro cause or genetic causes given it commonly runs in families, but no one is willing yet to push that, because ITW is too variable.
Response 4: We completely concur with your viewpoint that ITW is highly diverse, with numerous hypotheses and even verified factors for individual cases (such as heredity, presence of specific genes, muscle tone, muscle fiber condition, etc.) that categorize this condition among other well-defined diseases. Our working assumption is limited to instances of ITW where sensory-motor disorders are present.
Point 5: I’d urge is authors are committed to this theory, work on linking toe walking and autism and gut health prior to then stepping out to ITW, children with itw are too diverse a population and hypothesis too far from anything right now to be believed without creative more stress on parents.
Response 5: Given this information, our review aimed to organize the various facts related to the group of motor pathologies known as ITW. These conditions not only negatively impact children's quality of life but also affect their parents. Our attempt may be modest, but it is necessary to understand and address these issues. Perhaps we have different experiences with the Reviewer of communication with parents. In our experience, we have found that the main source of stress for parents is not us, but rather the insufficiency of effective treatment methods. To develop such methods, we need new and innovative ideas that challenge current theories and connect previously unrelated phenomena. Only by being open to these possibilities can we make progress in finding better solutions for families in need.
While we do not claim to have all the answers, it would be imprudent to ignore the possibility of the microbiome playing a role in some cases of ITW. Therefore, the review focused extensively on the dynamics of ITW formation, both in normal birth and cesarean section cases. It particularly highlighted that artificial feeding, as opposed to breastfeeding, contributes to the increased number of gait and TW disorders. With regard to work on linking toe walking and autism, researchers of the gut microbiome of children with ASD, whose work was reviewed by Plaza-Diaz et al, have access to personal data. They could find correlations between changes in the gut microbiome and toe walking, if they did such work. We can significantly enhance our comprehension of the factors that cause motor impairment by adopting a meticulous approach that considers the consistent findings pertaining to the diversity of the microbiome's configuration in various research studies.
We believe that a comprehensive approach to the study of ITW is necessary due to the varied nature of its manifestations. Furthermore, a thorough examination of metabolic disorders responsible for both TW and ITW is crucial.

Reviewer 2 Report
Well effort by authors in the previous revision. Some correction needed as follows.
1. Line 40, it is suggested to make only as “Psychomotor”.
2. Line 54, I think the authors made a mistake that write “5,5” that should be “5.5”.
3. Line 288, I feel the illustrated figure would be improved in coloured picture with three dimension model.
4. Based on Jamari et al., gait is consisting of three main components, there are acting load, rang of motion, and cycle. Please provide this information along with relevant reference as follows: https://doi.org/10.1016/j.heliyon.2022.e12050
-
Author Response
Point 1: Line 40, it is suggested to make only as “Psychomotor”.
Response 1: We agree with this edit and left only "Psychomotor".
Point 2: Line 54, I think the authors made a mistake that write “5,5” that should be “5.5”.
Response 2: Thank you very much, and we have fixed this typo.
Point 3: Line 288, I feel the illustrated figure would be improved in coloured picture with three dimension model.
Response 3: Admittedly, our artistic skills may leave something to be desired, nevertheless, after attempting to implement your suggestions, the result was even less satisfactory. Therefore, we have decided to leave the drawing unaltered.
Point 4: Based on Jamari et al., gait is consisting of three main components, there are acting load, rang of motion, and cycle. Please provide this information along with relevant reference as follows: https://doi.org/10.1016/j.heliyon.2022.e12050
Response 4: Thanks to your valuable suggestion, we were able to substantiate the effect of modified action load on muscle contracture by citing a relevant article on gait components (see Ref. 171, line 945-948)

Round 3
Reviewer 2 Report
Appreciate to the authors for their effort in this stage. Some remarks still needs to addressed by the authors as follows:
1. Line 109, Please do not use “we”, make it into passive.
2. Line 111, please give additional relevant reference regarding autism spectrum disorder as follows: https://doi.org/10.3390/bioengineering9020048
3. Line 566, recommended to make it as narrative, not point-by-point present form.
4. Related to previous review report in round 2 number 3, figure still encouraging to improved that makes from three dimensional illustrated to make it more realistic. But, if is not possible, it is okay.
-
Round 4
Reviewer 2 Report
I am recommended to accept the manuscript, thanks
-